# Accurate additive manufacturing of lightweight and elastic carbons using plastic precursors

Paul Smith[1], Jiayue Hu[2], Anthony Griffin [1], Mark Robertson[1], Alejandro Güillen Obando[1], Ethan Bounds[1], Carmen B. Dunn[1], Changhuai Ye[3], Ling Liu [2] ✉ & Zhe Qiang [2] ✉

Despite groundbreaking advances in the additive manufacturing of polymers, metals, and ceramics, scaled and accurate production of structured carbons remains largely underdeveloped. This work reports a simple method to produce complex carbon materials with very low dimensional shrinkage from printed to carbonized state (less than 4%), using commercially available polypropylene precursors and a fused filament fabrication-based process. The control of macrostructural retention is enabled by the inclusion of fiber fillers regardless of the crosslinking degree of the polypropylene matrix, providing a significant advantage to directly control the density, porosity, and mechanical properties of 3D printed carbons. Using the same printed plastic precursors, different mechanical responses of derived carbons can be obtained, notably from stiff to highly compressible. This report harnesses the power of additive manufacturing for producing carbons with accurately controlled structure and properties, while enabling great opportunities for various applications.

Additive manufacturing (AM) is a key technology of Industry 4.0 to produce customized parts with high resource efficiency and the unique advantage of integration with intelligent systems for smart material design and manufacturing[1]. Over the years, many AM-based technologies have been developed and penetrated various industrial sectors and commercial markets with continued interest in exploring their application domains[2,3]. Looking forward, enabling AM to access a vast material design space is extremely important and necessary. While tremendous development of AM has been made in various polymer, metal, and ceramic systems, its application to create carbon parts with controlled structure and functionalities remains largely underdeveloped. The effective absence of AM of carbons at commercial scale indeed draws sharp contrast with their indispensable roles in catalysis[4], energy storage[5], biomedicine[6], high-performance composites[7], as well as many other important areas. Addressing this significant gap will transform the future of the AM industry, while

unlocking the enormous potential of carbons in various application domains.

To date, most reported systems for AM of carbons either rely on the extrusion of slurries containing carbon additives, followed by matrix/solvent removal[8], or the printing of polymeric carbon precursors, with subsequent steps of crosslinking and pyrolysis[9–12]. Specifically, direct ink writing (DIW) can be used to selectively deposit a slurry of dispersed functional carbon nanomaterials in a water-based or polymer-based system, such as carbon nanotubes, graphene oxide (GO), or reduced graphene oxide (rGO)[13,14]. The rheological profile of these inks can be adjusted for enabling proper printability to allow creation of complex structures. The solvent/polymer matrix is then removed through supercritical drying or pyrolysis, leading to carbon-only structures[15]. While DIW has also been developed for printing cellulose/lignosulfonate-based materials serving as carbon precursors[16], most traditional polymeric carbon precursors such as

[1]School of Polymer Science and Engineering, The University of Southern Mississippi, 118 College Drive, Hattiesburg, MS 39406, USA. [2]Department of Mechanical Engineering, Temple University, 1801N Broad Street, Philadelphia, PA 19122, USA. [3]State Key Laboratory for Modification of Chemical Fibers and Polymer Materials, College of Materials Science and Engineering, Donghua University, Shanghai 201620, China. ✉e-mail: ling.liu@temple.edu; zhe.qiang@usm.edu

polyacrylonitrile and polyimides are difficult to process through extrusion-based printing methods due to their high melting temperature and limited solubility[17]. Alternatively, UV-assisted AM technologies can be employed to achieve complex 3D structures of carbon precursors. These techniques have been explored for a variety of different precursor systems such as mimosa tannins in an acrylate-based UV curable resin at high loading levels[18]. Once printed and post-cured, parts containing high-char yielding mimosa tannins can undergo pyrolysis with a carbon yield of up to 23% and dimensional shrinkage of ~26%. In the case of polyimides, AM has been demonstrated through vat photopolymerization as well as UV-assisted DIW to form precursor scaffolds[9]. The printed structures were then crosslinked and pyrolyzed to yield structured carbon products, resulting an isotropic dimensional shrinkage of ~55% upon converting printed to carbonized parts. More recently, AM of carbons were demonstrated through using commodity polypropylene (PP) as precursors via two steps of sulfonation-induced crosslinking and carbonization, creating structured carbons with robust mechanical performance[19]. This approach employs low-cost precursors and simple processing conditions, however, printed PP parts experience anisotropic dimensional shrinkage (up to 20%) after pyrolysis. While this degree of shrinkage is lower than previous reports using most other carbon precursors, further development for enabling accurate dimensional control over the final products is necessary for practical implementation.

Furthermore, creating carbon parts with controlled compressible/elastic mechanical responses represents a strong need for many emerging technological applications, including but not limited to wearable strain-sensing electronics[20] and deformable electronic skins[21]. Over the past years, a variety of approaches have been demonstrated to prepare compressible carbon materials, primarily through intrinsic pore engineering within framework and/or design of material architectures to enable mechanical metamaterials[12,22–25], where, notably, AM is uniquely capable to create complex material topology. Specifically, in-situ self-assembly[26,27], freeze-casting[28], templated growth[29] and vapor deposition[30] have been demonstrated to create lightweight and highly porous carbons, where the reversible material deformation can be attributed to the presence of void spaces allowing for efficient stress de-concentration[31]. Alternatively, previous work also employed direct laser writing and pyrolysis to create lightweight and compressible carbon lattices[32–34]. However, these methods require the use of special precursors (often not commercially available) and high-cost instruments, and have an unavoidable high degree of volumetric shrinkage (>90%) upon conversion to carbons, which largely limits their resource efficiency toward scaling up.

In this work, we report an AM system to on-demand create carbon parts by using commercially available PP filaments containing carbon fiber fillers (PP-CF). After sulfonation-enabled crosslinking, FFF printed PP-CF parts can be pyrolyzed to create CF-reinforced carbon composites. Noteworthily, the presence of CF in the printed parts significantly limits the shrinkage of the polymer precursor matrix during its transformation to carbon. It is found that the degree of shrinkage reduces with increasing CF content, reaching <4% with 15 wt% CF content, which is independent on the crosslinking degree of the PP matrix. By leveraging this advantage, lightweight porous carbons containing complex geometries can be successfully prepared with tunable material density, which are directly controlled by PP crosslinking time. As a result, altered mechanical properties of PP-CF derived carbon parts, from stiff to soft and compressible, can be attained from the same printed specimens through simply varying the processing conditions. Particularly, the macropores within the carbon parts, introduced through under-crosslinking of PP precursors, enable high compressibility and elasticity with >50% of maximum strain deformation. The reported approach is simple and scalable, representing great potential for their transformative impact to enable AM of carbons at scale with control over material structure, density, and mechanical performance.

## Results

### Sulfonation-induced crosslinking of FFF-printed PP-CF precursors

This work reports a robust AM system to create structured carbons with high dimensional control from printed to carbonized states, using low-cost and commercially available 3D printing filaments as starting materials. As shown in Fig. 1a, a polypropylene-based filament, containing 15wt% chopped carbon fiber fillers (PP-CF, Supplementary Figs. 1, 2), can be printed using the fused filament fabrication (FFF) method, and then submerged in concentrated sulfuric acid at elevated temperatures for crosslinking of PP matrix. 150 °C was selected as the sulfonation temperature for the model investigation. The use of sulfuric acid requires extra cautious for reaction safety and waste management; a relevant work demonstrated a continuous multiphase reactor for polyolefin sulfonation and crosslinking toward carbon fiber production[35]. After crosslinking, parts can be carbonized at 800 °C under an inert atmosphere to produce structured carbon products. While this manufacturing process is similar to a previous report[19], a significant difference was observed in the final structure of resulting carbons. Particularly, the model gyroid-shape carbon sample (~15 mm in all dimensions) exhibits very limited structural change upon conversion from as-printed to carbonized states, which is drastically different from the previous study showing ~20% of dimensional shrinkage along the printing directions (in-plane). Such accurate control over the macroscopic structure of final carbon products, simply enabled by the inclusion of CF fillers in the precursor filaments, represents a critical technological breakthrough in AM of carbon.

We first investigated the impact of CF on the crosslinking kinetics of the PP matrix throughout the sulfonation reaction, using a model system of gyroid-shaped cubic structures (~15 mm in all dimensions) printed from PP-CF with a wall thickness of 0.6 mm. To briefly describe the crosslinking mechanism of PP precursors, sulfuric acid can functionalize PP chains at elevated temperatures, followed by the dissociations of sulfonyl groups that result in alkene bonds within polymer backbones. Subsequently, these unsaturated groups from the sulfonation reaction can continue to react and eventually form crosslinked networks through intermolecular radical couplings; the crosslinked PP is an effective carbon precursor[36,37]. Fig. 1b shows the material mass gain (due to the addition of bulky sulfur-containing functional groups into PP backbones) and the change in PP crystallinity (due to the transformation from linear polymer to a crosslinked system) as a function of crosslinking time at 150 °C. Limited changes in both values were observed within the first 2 h, while by extending reaction time to 4 h, a sharp increase in the mass of PP-CF parts and decrease in PP crystallinity were found as a function of extending reaction time (Supplementary Fig. 3). After 12 h, changes in crystallinity reach a value of 0%, indicating that PP becomes completely amorphous, while mass gain continues to increase until 24 h. These results of indicating PP-CF crosslinking kinetics are consistent with gel fraction measurements (Supplementary Fig. 4) and Fourier transform infrared spectroscopy (FTIR) data (Supplementary Fig. 5); a simplified scheme illustrating chemical conversion of PP to crosslinked carbon precursor is also provided in Supplementary Fig. 5a. Briefly, changes in characteristic bands in the FTIR spectra of sulfonated PP-CF sample indicate the decrease in olefinic character through the reduction in the intensity of C-H bands at 2920 $cm^{-1}$ and 1600 $cm^{-1}$, the increase in characteristic -OH stretching band at 3300 $cm^{-1}$, as well as an increase in the bands at 1250 $cm^{-1}$ and 1000 $cm^{-1}$, which are associated with the functionalization of sulfonic acid groups to the PP backbone. Additionally, the evolution of bands at 1650 $cm^{-1}$ corresponds to the installation of double bonds (crosslinking sites) in the PP backbone from the departure of sulfonic acid groups. We previously found that crosslinking of 3D printed, thick PP samples involves an important cracking-facilitated reaction mechanism associated with spatial heterogeneity in the degree of PP sulfonation within printed layers.

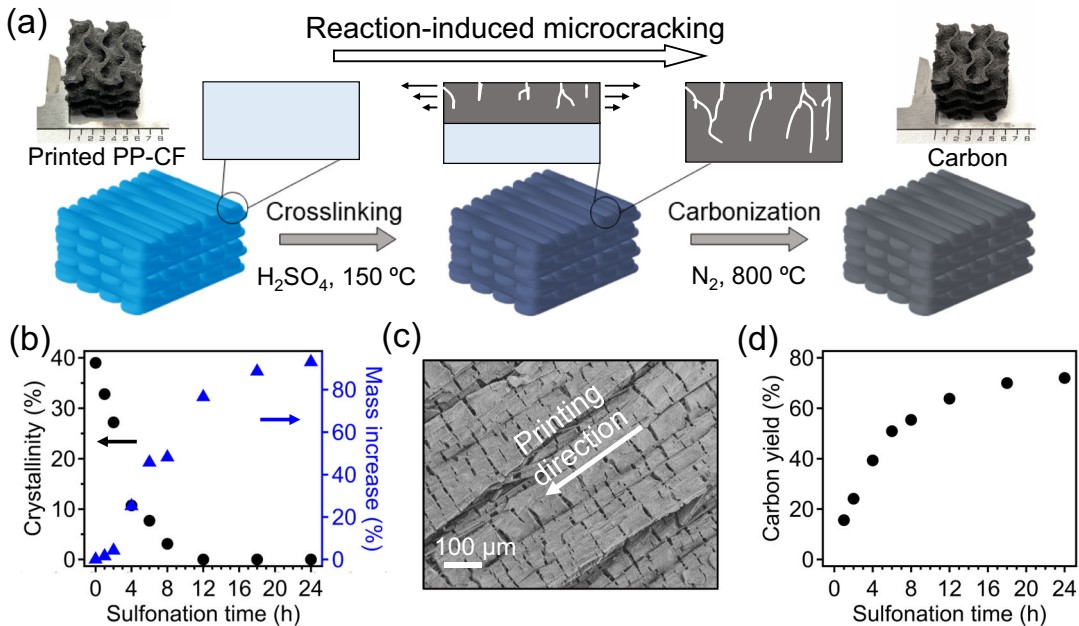

**Fig. 1 | Crosslinking and carbonization of FFF-printed PP-CF. a** Processing scheme for AM of carbons using commercial PP-CF as the precursor. Note the size of gyroid model shown in the inset photos is 0.6 inch (~15 mm). **b** Mass gain and degree of crystallinity of PP-CF as a function of sulfonation time at 150 °C. **c** SEM image of PP-CF after crosslinking for 2 h, and directional cracks normal to the printing direction were observed. **d** Carbon yield of PP-CF precursors as a function of sulfonation time.

Specifically, since the sulfonation reaction is through diffusion and from the outside in, the crosslinked outer layer can first become hydrophilic, allowing for potential swelling when immersed in acid, while the inner layers of pristine PP remain hydrophobic. This leads to mismatched expansion of distinct portions within PP layers and create stress for generating microcracks. As shown in Fig. 1c and Supplementary Fig. 6a–d, cracks were observed after 2 h of reaction time, which is much earlier than their PP counterparts with the absence of fillers (~4 h, Supplementary Fig. 6e–h) under identical crosslinking conditions (i.e. 150 °C). This result also explains why the total amount of time required for PP-CF to reach a fully crosslinked state (~18 h) is significantly less than PP filaments with the absence of CF (~48 h). The directionality of these microcracks in PP-CF systems can be attributed to the anisotropically enhanced mechanical properties in printed parts due to the presence of CFs (Supplementary Fig. 7), which were aligned by the extrusion shear force involved during the FFF process along with the printing direction (Supplementary Fig. 8). Note that the alignment of these micro-cracks also confirms that their formation mechanism is associated with sulfonation-induced mismatched stress, rather than the release of gaseous byproducts from PP crosslinking reaction. Upon pyrolysis, ~67 wt% carbon yield, compared with the initial printed parts, was achieved for samples crosslinked for 18 h and longer. Considering the presence of 15 wt% CF in the system, the actual carbon yield from PP precursor is ~61 wt%, which is similar to previous reports of PP-derived carbons[38–40]. Thermogravimetric analysis-mass spectrometry confirms the presence of gaseous product during pyrolysis, including $SO_2$, $CO_2$ and CO molecules (Supplementary Fig. 9). Additionally, completion of carbonization at 800 °C was confirmed through FTIR spectroscopy (Supplementary Fig. 10); no distinct bands were observed due to the absence of functional groups.

**Structured carbons from pyrolysis of crosslinked PP-CF precursors**

A suite of characterization methods was employed to understand the physical properties of the PP-CF derived carbon, which was crosslinked at 150 °C for 18 h, including their microstructures, pore textures and porosity, as well as mechanical performance. As shown in Fig. 2a, the

resulting carbons exhibit characteristic unidirectional micro-cracks, which were generated during sulfonation and retained upon pyrolysis. It was found that the averaged distance between these microcracks was slightly reduced from 85 μm (after crosslinking) to 82 μm (after carbonization), which is consistent with the observation of minimal dimensional shrinkage of printed parts after carbonization. Noteworthily, higher magnification SEM imaging of these carbons (Fig. 2b) revealed the clear presence of microporous structures with a broad distribution of pore sizes in the continuous carbon framework. The formation of pores in these PP-CF derived carbons is anticipated considering the loss of mass upon pyrolysis coupled with near-zero dimensional shrinkage, leading to a framework density of ~0.7 g/cm³. The presence of CFs physically restricts densification of the crosslinked PP matrix during conversion to carbon, which encourages the formation of void spaces. We note that these micropores within carbon frameworks from this particular sample are not resulted from degradation of potential uncrosslinked portions within PP-CF filaments, which will be discussed in the following sections.

To further understand the porous nature of carbons derived from crosslinked PP-CF parts, physisorption measurements were employed using liquid nitrogen as a probe, and the resulting adsorption and desorption isotherms are shown in Fig. 2c. A typical type II isotherm is observed, indicating the mixed presence of micro- and macro-pores in the carbon framework. Specifically, the material exhibits a BET surface area of 302 m²/g and a micropore volume of 0.2 cm³/g, while its mesopore size distribution is shown in Supplementary Fig. 11. Additionally, Raman spectroscopy was used to understand the graphitization degree of PP-CF derived carbons, and two distinct and relatively broad peaks associated with disordered and graphitic structures located at ~1350 cm⁻¹ and ~1580 cm⁻¹ can be observed in Fig. 2d. The ratio of the intensity of these peaks was found to be $I_D/I_G = 1.25$, indicating that these carbons are amorphous in nature, which is consistent with other studies of polyolefin-derived functional carbons[41].

Compressive mechanical testing (normal to the Z-direction of the printed specimen (Supplementary Fig. 12) was performed on structured carbons derived from PP-CF crosslinked for 18 h, which had an in-fill density of 40% from FFF printing; a representative stress-strain

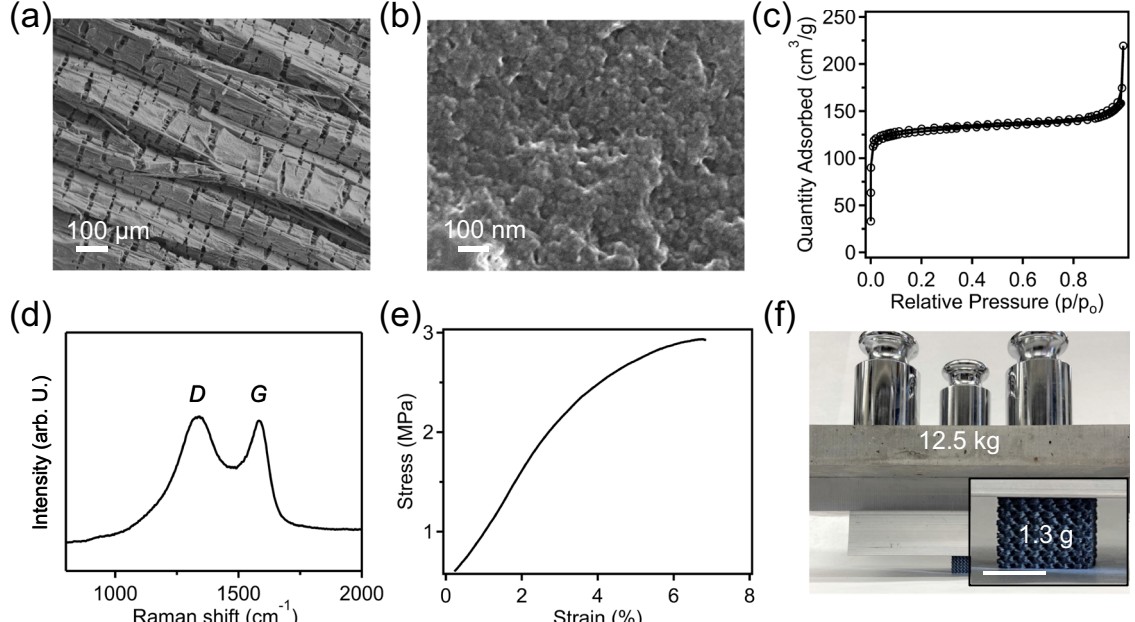

**Fig. 2 | Properties of 3D-printed PP-CF derived carbons. a** SEM image of unidirectional cracks in PP-CF derived carbon, which was sulfonated at 150 °C for 18 h. **b** High magnification SEM image of PP-CF derived carbon showing its porous nature at nanoscale. **c** BET isotherm of PP-CF derived carbon, which was sulfonated for 18 h. **d** Raman spectra of carbon derived from crosslinked PP-CF precursors, including disordered (D) and graphitic (G) peaks, with an intensity ratio ($I_D/I_G$) of 1.25. **e** Representative compressive stress-strain curve of PP-CF derived carbon (crosslinked at 150 °C for 18 h) with cubic gyroid structure and a 40% in-fill density. **f** 1.3 g printed carbon structure can hold at least 12.5 kg of weights, scale bar: 1 cm.

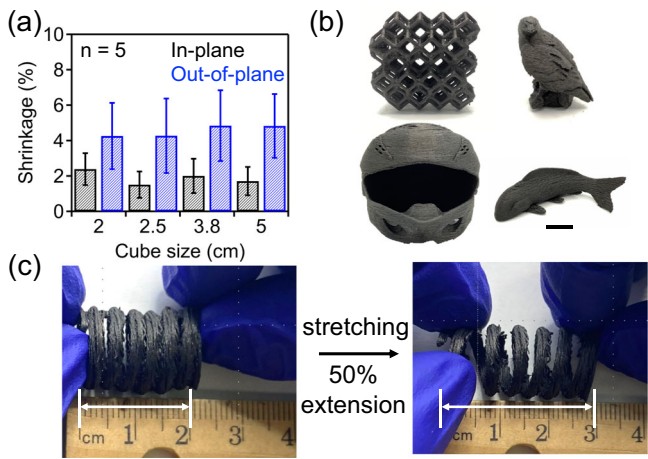

**Fig. 3 | Low dimensional shrinkage of PP-CF upon carbonization. a** Dimensional shrinkage of PP-CF gyroid cubes upon conversion to carbon products. In-plane shrinkage is reported in black bars on the left, out-of-plane shrinkage is given as blue bars on the right. All parts were crosslinked for 18 h at 150 °C. Values reported are an average of 5 samples with error bars representing the standard deviation. **b** Complex carbon structures prepared from PP-CF precursors using the reported method, scale bar: 2 cm. **c** Deformability of a carbon spring derived from PP-CF showing up to 50% extension without fracture.

curve of these samples is shown in Fig. 2e. These materials show an average elastic modulus of $63.8 \pm 9.3$ MPa and an ultimate strength of $2.9 \pm 0.2$ MPa, where failure occurred at 7% strain. Both the elastic modulus and ultimate strength of PP-CF derived carbon show a significant increase in compared to their counterparts of carbons derived from unfilled PP filaments (modulus $3.4 \pm 0.3$ MPa, ultimate strength $0.3 \pm 0.1$ MPa, Supplementary Fig. 13). We attribute this difference to the inclusion of high strength, high modulus CF within the PP matrix leading to mechanically stronger carbon composites. Through

normalizing the mechanical properties with their respective apparent density, it was found that PP-CF derived samples displayed a specific modulus of $235 \pm 34$ MPa·cm$^3$/g and a specific strength of $11 \pm 0.7$ MPa·cm$^3$/g, and these values are significantly increased compared to PP-derived carbons with the same geometry, which are $13.1 \pm 1.6$ MPa·cm$^3$/g and $0.9 \pm 0.3$ MPa·cm$^3$/g, respectively. Additionally, for the mechanical properties of carbon along the out-of-plane printing direction, while PP-derived carbon resulted in a compressive strength of $0.3 \pm 0.1$ MPa and an elastic modulus of $3.3 \pm 0.2$ MPa, PP-CF derived carbon had an increased strength of $0.77 \pm 0.1$ MPa and modulus of $37.1 \pm 8.2$ MPa. Moreover, an example to visualize the strong mechanical properties of these lightweight carbons is provided in Fig. 2f; a 1.3 g gyroid structured carbon was able to easily support 12.5 kg of mass, indicating a strength-to-weight ratio of at least 9600:1.

Dimensional shrinkage of parts upon polymer-to-carbon conversion, process scalability, and reproducibility were quantitatively assessed in our work; a series of cube shaped PP-CF parts were prepared, with dimensions ranging from 2 to 5 cm and an in-fill density of 40%. After carbonization, all samples exhibited only an average of 2% shrinkage in the in-plane directions (X and Y directions, along FFF deposition direction, Supplementary Fig. 12) and an average of 4% in the out-of-plane direction (Z-direction, normal to deposition direction), relative to the as-printed PP-CF parts (Fig. 3a). For comparison, converting neat PP (no fillers) to carbons results in an averaged dimensional shrinkage of 22% and 9% along in-printing plane and out-of-printing plane direction, respectively; X and Y directions do not show any statistically noticeable difference in the degree of shrinkage for all samples from PP-CF and PP precursors. Additionally, we found that the degree of dimensional shrinkage (from PP-CF to carbon) along in-plane printing direction did not change even when increasing carbonization temperature to 1400 °C, while the out-of-plane shrinkage increased to ~8% (Supplementary Fig. 14); the slight anisotropic shrinkage can be possibility due to the nature of FFF printed parts or the templating effect from the CFs which are aligned with the in-plane direction. As shown in Supplementary Fig. 15, after carbonization the alignment of CF was completely retained in the final carbon

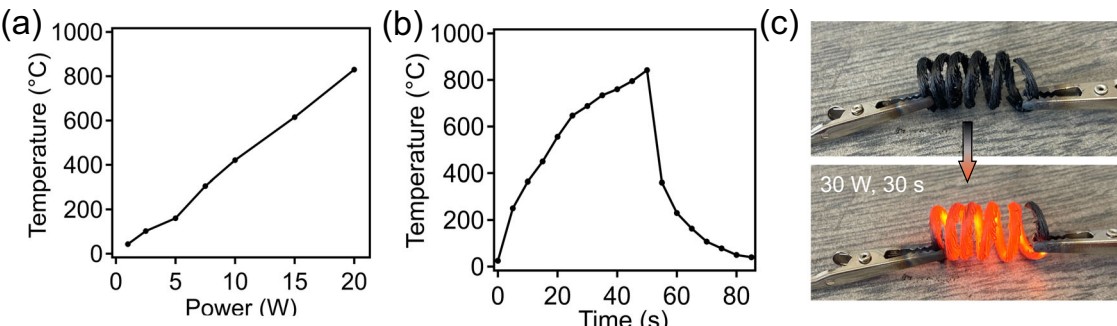

**Fig. 4 | Joule heating performance of 3D printed PP-CF derived carbons. a** Joule heating performance of PP-CF derived carbon as a function of supplied power. **b** Temperature as a function of time held at a supplied power of 20 W. **c** Joule heating performance of PP-CF derived carbon spring after 30 s under 30 W.

composites. We attribute the very low dimensional shrinkage of our samples to the presence of CF in the printing filaments, which restricts the volumetric contraction of polymer matrix during their conversion to carbonaceous products. Similar filler impacts on significantly reducing volume shrinkage of precursors upon carbonization were also observed in several previous studies[16,42,43]. For example, Liao et al. found <0.5% dimensional shrinkage for preparing carbon/carbon composites through laser bade powder bed fusion, which were carbonized from crosslinked phenolic resin precursors containing up to 60 wt% carbon fibers[44]. Wang et al. reported that dimensional shrinkage of 3D printed parts upon converting CF-reinforced polymer inks to ceramic composites is highly dependent on filler loading content; the inclusion of 50 wt% CF fillers in the polymer precursors leads to 0.4% shrinkage[45]. Another analog for explaining the excellent dimensional retention of our system enabled by CF inclusion can be drawn from a recent work where the influence of high carbon nanotube loadings was shown to result in very low volumetric shrinkage (<10%) even after the removal of polymer matrix removal. However, this system requires a high amount of sacrificial polymer for enabling 3D printing of carbon fillers, which could limit the resource efficiency[46].

To further confirm the role of CF on enhancing the macroscopic structure retention upon PP to carbon conversion, a series of PP-based FFF filaments were prepared containing different CF loading content from 0 to 10 wt%. These filaments were printed into identical specimens and their shrinkage behaviors were examined (Supplementary Fig. 16). A clear trend was observed as increasing fiber loading led to reduced dimensional shrinkage. Specifically, when the CF content increased from 2.5% to 5%, in-plane shrinkage was reduced from 16–11%. For PP containing 10 wt% CF, 3% shrinkage along the in-plane direction and 5% along the out-of-plane direction were observed, similar to results from PP-CF in Fig. 2a. Extending this system to another commercially available fiber-reinforced filaments, PP-GF (PP containing 30 wt% glass fibers), leads to consistent, very minimal dimension shrinkage behaviors upon carbonization at 800 °C (Supplementary Fig. 17), confirming the generalizability of our system design to create structured carbons with excellent structural retention throughout the entire manufacturing process.

Leveraging the unique structural complexity afforded by AM, we employed our process to prepare a variety of carbon materials with different complex geometries, including a rhombic dodecahedral lattice, a golden eagle, a helmet, and a koi. All parts successfully preserved their printed geometry with a consistent shrinkage of <5% across all directions and a carbon yield of >65 wt%, as shown in Fig. 3b. Detailed information about shrinkage and mass yield are included in Supplementary Table 1. These results further indicate that our simple and scalable method is broadly applicable to produce carbons of different sizes and geometries with consistently accurate structural control, while employing commercially available filaments as starting materials. Furthermore, a carbon spring (Fig. 3c) was fabricated which

can exhibit up to 50% deformation upon stretching, demonstrating that carbons derived from 3D printed PP-CF are mechanically strong, indicating our method may provide a feasible pathway to manufacture carbons with topology-controlled mechanical performance.

An excellent demonstration of the applicability of these 3D printed carbons from PP-CF filaments is for Joule heating, which can potentially provide an important material solution for industrial decarbonization. Specifically, according to Joule's law, increasing the electrical resistance of conductors can result in a higher temperature under the same current. The amorphous nature of these PP-derived carbons provides a significant advantage in the Joule heating capability compared to their counterparts of metals and more graphitic carbon-based[47,48], which are typically much more electrically conductive. As shown in Fig. 4a, by increasing the input power from 5 W to 10 W and 20 W, the carbon spring can increase temperature from 186 °C to 422 °C and 810 °C, respectively. The fast heating on and off response of these carbons is shown in Fig. 4b, which only 40 s is required for them reaching the maximum temperature with 20 W of power supply, while a rapid cooling from 800 °C to 200 °C occurs only within 20 s, indicating a full electrical power-controlled heating behavior. Figure 4c shows the efficient joule heating of a deformable/stretchable carbon spring exhibiting a glowing red color after only 30 s with using 30 W of supplied power, further suggesting their great potential in Joule-heating applications with very low energy consumption. In conjugation with high thermal stability of carbons, this feature is particularly desired for the electrification of high temperature heating processes toward a carbon-neutral economy[49,50].

## Controlling density and mechanical performance of structured carbons through varying degree of crosslinking

As previously discussed, the macroscopic structural retention of printed samples during polymer-to-carbon conversion is controlled by the inclusion of fibers and their loading content, which provides a unique advantage in our system to allow process-tunable density from printed PP-CF parts through varying crosslinking time. Specifically, as shown in Fig. 5a, changing sulfonation time from 4 h to 24 h leads to consistent in-plane dimensional shrinkage of <3%. However, due to their substantial differences in the crosslinking degree of PP matrix, carbon yield is significantly different (Supplementary Fig. 18), resulting in a broad range of framework density varying from 0.20 g/cm³ to 0.75 g/cm³. From these results, it was found that PP-CF with 4 h sulfonation time at 150 °C has a very high porosity of 91%, which decreases to 84% and 7% and 63% by extending reaction time from 6 h to 8 h and 12 h, respectively. Further increasing the reaction time to 18 h and 24 h only leads to slight reductions in porosity, which is consistent with our carbon yield results in Fig. 1d and Supplementary Fig. 18. As discussed in the previous section, ~60% porosity from fully crosslinked PP-CF samples is all attributed to the presence of micropores (Fig. 2b), which are formed from limited volumetric shrinkage of

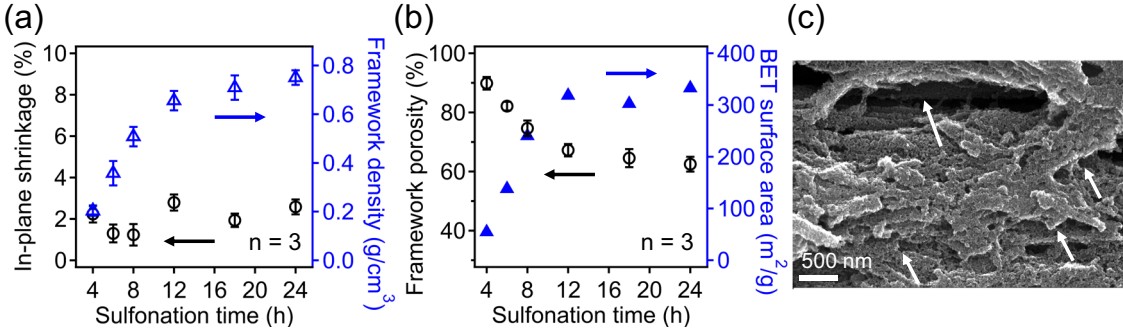

**Fig. 5 | Controlling porosity and pore textures via sulfonation time. a** In-plane dimensional shrinkage and porosity of PP-CF derived carbon as a function of sulfonation time reported for a samples size of 3, error bars represent one standard deviation. **b** Framework porosity and BET surface area of carbonized samples as a function of increasing sulfonation time. Framework porosity is represented by an average of 3 measurements with error bars representing the standard deviation. BET surface areas are reported for a single sample. **c** Representative SEM image of 8 h sulfonated PP-CF after carbonization showing characteristic void spaces left by the removal of insufficiently crosslinked domains.

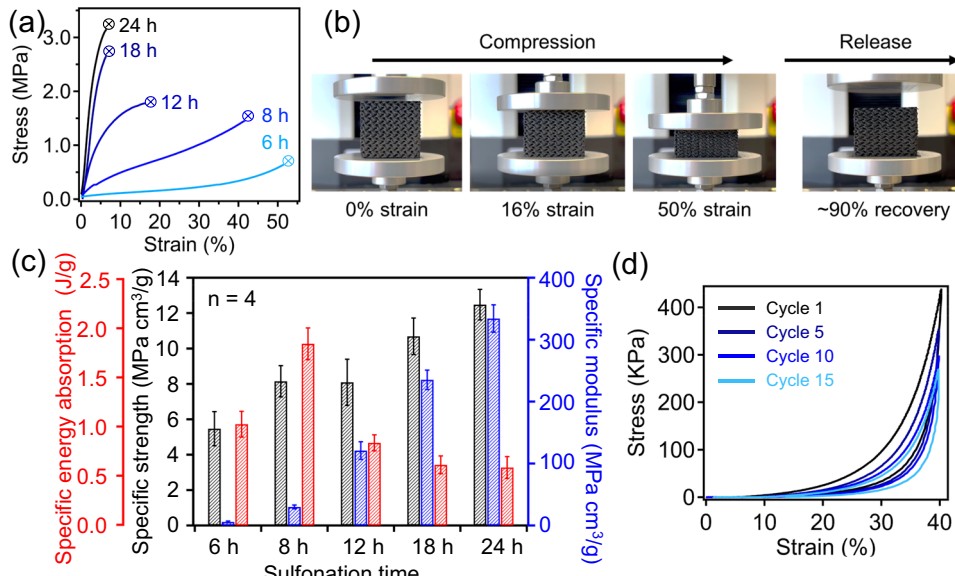

**Fig. 6 | Controlling mechanical property via varying sulfonation time.**
**a** Representative compressive stress-strain curves of PP-CF derived carbons, which were crosslinked at 150 °C for different times. **b** Photos showing compressive strain recovery behavior in carbons derived from partially crosslinked PP-CF. **c** Resulting specific mechanical properties of carbons including compressive strength, modulus, and energy sorption with varying crosslinking time. Reported values are averages based on four measurements, and error bars represent the standard deviation. **d** Cyclic compressive stress-strain curves of carbonized samples (crosslinked at 150 °C for 6 h) brought to 40% strain.

samples during carbonization. Furthermore, degree of crosslinking of PP precursor can also have a strong influence on the BET surface of resulting carbons, increasing from 54 m²/g from 4 h of sulfonation to a plateau value of ~320 m²/g for samples crosslinked for 12 h and longer. The liquid nitrogen sorption isotherms of these samples are included in Supplementary Fig. 19. SEM images in Fig. 5c and Supplementary Figs. 20, 21 confirm the presence of macropores (with varied sizes from hundreds of nanometers to more than several micrometers) within the framework of carbons that are derived from under-crosslinked samples, including crosslinking time of 4 h, 6 h and 8 h. Collectively, these results show that our method provides a facile handle to form porous carbon framework with direct control over the density and porosity by simply varying the reaction time of the same printed parts.

The changes in porous structures of samples as well as their framework density often strongly impact the material mechanical properties. Figure 6a shows the compression stress-strain behaviors of the resulting materials with varied sulfonation time; a clear trend can be observed where lower degrees of crosslinking in PP-CF parts leads to reduced modulus and increased compressibility. For the porous structured carbon prepared by only crosslinking for 6 h, a maximun strain of 54% can be obtained under 550 KPa stress; such highly compressible material behavior is drastically different from hard and stiff nature of carbon framework, which can be attributed to their high porosity. Noteworthily, using the same printed PP-CF parts, simply increasing the crosslinking reaction time from 6 h to 24 h can lead to carbons with very different mechancial behaviors, including signficantly stiffer and less compressible framework. Direct control over mechanical properties from varying PP matrix crosslinking degree also confirms that the elastic and compressible behaviors of our 3D-printed carbons are attributed to the instrinsic pore structures, which are resulted from selective removal of uncrosslinked portions of polymer matrix rather than their macroscopic geometry or topology. Additionally, Fig. 6b shows photos of a sample (crosslinked for 6 h) under compressive mechanical testing, with a high strain recovery at 90% (Supplementary Movie 1). The quantitative mechanical properties of PP-CF derived carbon samples from different crosslinking time are

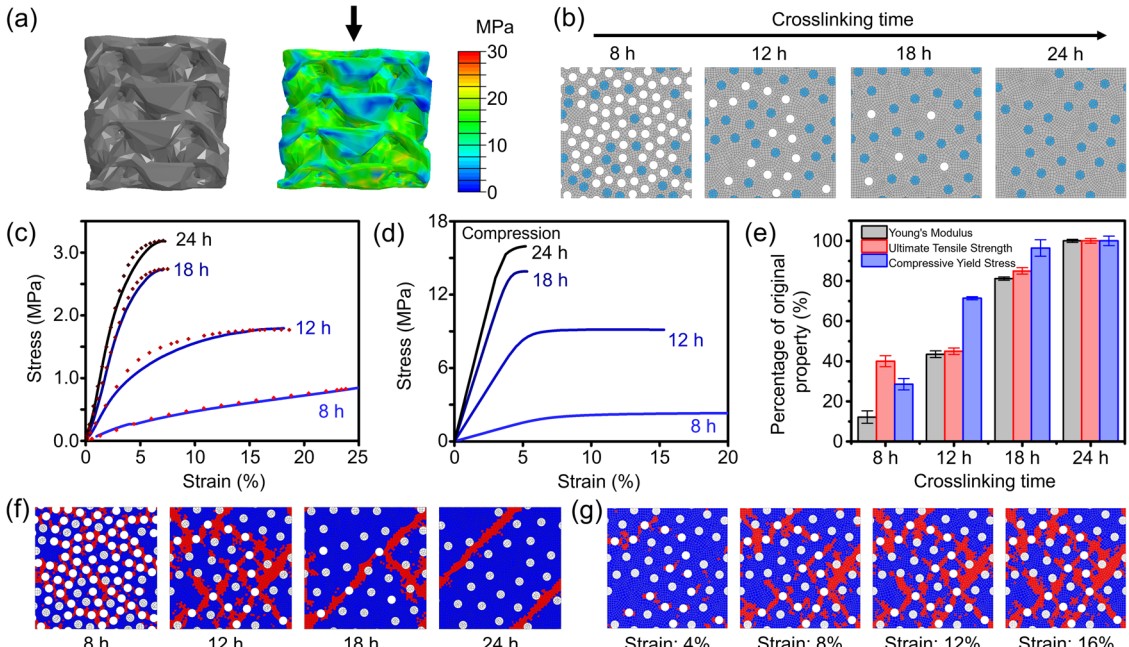

**Fig. 7 | Simulations for structure-property relationship of PP-CF dervied carbons. a** Compression of a 40% in-fill density gyroid structure (left: finite element modeling (FEM) simulation model; right: von Mises stress in the deformed structure with a global strain of 10%). **b** RVE models for varied crosslinking time (gray: matrix; blue: carbon fibers; white: macropores). Porosity from left to right: 28.4% (8 h); 5.8% (12 h); 1.9% (18 h); and 0% (24 h). **c** Compressive stress-strain curves of the gyroid structure (solid lines: experiment; dotted lines: FEM simulation).

**d** Compressive stress-strain curves of the RVE microstructures. **e** Mechanical properties of the PP-derived matrix versus the crosslinking time. Values reported are an average of 5 samples with error bars representing one standard deviation. **f** Failure pattern in the RVE under compression along the vertical direction (red elements are damaged). **g** Failure progression in the 12 h sample with the compressive strain ranging from 4% to 16%.

provided in Fig. 6c. In general, the specific strength and modulus of these materials were observed to increase with longer sulfonation times, which can be attributed to the corresponding reduction in porosity as further crosslinking is achieved. These values reached a maximum in both modulus and strength after 24 h of sulfonation with a specific modulus of 334 MPa·cm$^3$/g and a specfic strength of 12.5 MPa·cm$^3$/g. In the case of specific energy absorption it was found that sample sulfonated for 8 h (and thus a partially crosslinked state) was able to achieve a high specific energy absorption of 1.8 MJ/m$^3$·cm$^3$/g. The ability of these PP-CF derived carbons to recover from compressive strain was further studied by cyclic compressive testing at 20% strain and 40% strain for 10 to 15 cycles (Supplementary Figs. 22 and Fig. 6d respectively, sample was crosslinked for 6 h at 150 °C). After initial plastic deformation occurred, the following cycles showed very similar mechanical properties with a slight reduction in modulus and peak load with increasing number of cycles. Briefly, cyclic compression to 40% strain resulted in a reduction in modulus from 750 KPa to a modulus of 720 KPa after 15 cycles, representing a retention of 96% elastic modulus while ultimate strength achieved ~60% property retention after 15 cycles. The reduced mechanical properties as a function of cycle number can be attributed to the partially irreversible damage accumulation, which is often observed in other studies of developing porous carbon framework[23,51,52]. Nevertherless, this mechanical property retention is comparable to and/or better than other works such as compressible aerogels[53].

To further understand the origin of elastic and compressible mechanical properties of these 3D printed carbon samples, simulation work was performed using a multiscale approach, encompassing three distinct length scales: (1) the macroscale featuring 3D-printed gyroid structures; (2) the mesoscale featuring composite microstructures with carbon fibers and macropores embedded in the matrix; and (3) the microscale featuring the carbon matrix incorporating

micropores. The macrostructure was modeled to have an edge length of 5 mm and 40% in-fill density as shown in Fig. 7a. At the mesoscale, the composite representative volume element (RVE) comprises randomly distributed fibers and macropores embedded in the matrix as illustrated in Fig. 7b. The porosity is strongly correlated with the crosslinking reaction and therefore, four RVE models were generated for different crosslinking times with the pore volume fractions inferred from experimental results. For simplicity, the matrix was modeled as a continuous phase with smeared properties taking into account the influence of micropores.

The results obtained from simulations demonstrate the significant impact of the macroporous structure (controlled by varied crosslinking time) on the mechanical properties of PP-CF derived carbons. Incorporating the reaction time-dependent mechanical properties of PP-CF derived carbons, the predictions of macroscale gyroid responses exhibit excellent agreement with experimental results as shown in Fig. 7c, confirming the accuracy of our multi-scale modeling approach. Note that the sample with 6 h sulfonation time exhibits a large strain exceeding 50%, resulting in large-scale surface contact within the gyroid structure. This case is therefore not included in the discussion, which primarily focuses on the samples with comparable deformation mechanisms. The modulus is extracted to be 11 MPa for the 6 h case, which is the lowest among all samples under investigation.

In addition to accurately describing experimental observations, the multiscale computational framework was employed to deduce cross-scale mechanical properties of the PP-CF derived carbon samples. At the mesoscale, Fig. 7d shows the compressive stress-strain curves of four carbon sample RVEs obtained from different crosslinking times. Note that the material properties presented here exclusively pertain to the sample without considering specific macrostructural forms (i.e. the gyroid topology), therefore representing base properties of the composite. The presence of dense macropores

at lower sulfonation times is shown to profoundly reduce both the modulus and strength of the composite. Specifically, elastic moduli of the carbon samples with the sulfonation time of 24 h, 18 h, 12 h, and 8 h are found to be 450 MPa, 350 MPa, 175 MPa, and 35 MPa, respectively; the compressive strength drops from 16 MPa to 14 MPa, 9 MPa, and 2 MPa. Overall, the PP-CF derived carbon sample with 8 h sulfonation exhibits very soft responses, while the composites with longer sulfonation are significantly harder. In parallel, the tensile properties of the RVEs can be found in Supplementary Fig. 23. Furthermore, simulation was performed to understand the impact of fiber/matrix interfacial strength on the mechanical properties of derived 3D printed carbon composites; this is for informing future technology improvement to allow design of products with enhanced mechanical performance. Although we recognized that pre-treatment of fiber surface prior to filament preparation, printing, and crosslinking may have a limited impact as the sulfonation reaction may lead to altered functionality, opportunities may exist to engineer fiber surface after crosslinking. In this work, we selected the RVE of sample that was crosslinked for 24 h as a representative model for investigation. The material properties of the CF-carbon matrix interface were varied in the range from 50% (weakened) to 150% (strengthened). As shown in Supplementary Fig. 24, it was found that the strength of the CF-carbon matrix interface has very limited impact on the compressive mechanical behaviors; however, with stronger interfacial interactions, the yield point of composite under tension is delayed, and a higher strength can be obtained.

Furthermore, at the microscale, Fig. 7e compares the mechanical properties of the carbon matrices prepared with different sulfonation times. These values represent the bulk properties of solid samples of PP-CF derived. Increasing the sulfonation time almost linearly increases the elastic modulus of the PP-derived matrix. 24 h crosslinking results in a completely crosslinked state, further leading to stiff PP-derived carbon with high yield and failure strengths. It is found that insufficient reaction time, such as 8 h, decreases the elastic modulus by ~90%, in addition to 60% reduction in the ultimate tensile strength and 70% reduction in the yield strength. The reduction in mechanical properties can be attributed to the presence of macropores produced from the removal of uncrosslinked PP parts upon carbonization.

The excellent compressibility achieved from the 3D printed carbon samples prepared with short sulfonation time can be attributed to two characteristics of the underlying materials: softness of PP-derived carbon matrices and non-local damage progression. Figure 7f illustrates compressive damage in the composite RVEs with varied macroporosities. It is clear that with decreasing crosslinking time, the damage zone spreads out across the entire RVE, rather than being localized as observed in the samples with 24 h and 18 h sulfonation time. The non-local damage alleviates stress concentrations and delays damage growth, leading to extended compressibility. After all, damage inception in the carbon originates from the fiber/matrix interfacial failure occurring especially in the vicinity of voids. Figure 7g shows such damage progression when the compressive strain is increased from 4% to 16% in the 12 h sample. In general, when a large amount of macrovoids are present, stress is less concentrated which delays the failure to occur. With reducing the amount of macropores, stress is more concentrated at the relatively scarce weak points (i.e., voids), thus quickly forming a discernible path of damage propagation. In parallel, failure mechanisms under tension can be found in Supplementary Fig. 23a. These findings provide valuable insights into the mechanical behavior of porous PP-CF-derived carbon composites, elucidating the critical role of sulfonation time in tailoring material properties and microstructure performance.

It is important to note that the mechanical strength and elastic modulus of these PP-CF derived carbons are lower than common carbon-based materials (such as polyacrylonitrile derived carbon fibers), which are collectively due to several reasons. First, the presence of CF in the crosslinked PP matrix limits the volume shrinkage of

sample upon high temperature carbonization, leading to the formation of micropores within carbon matrix, which these micropores could result in reduced mechanical properties. Second, the crosslinking process inherently generates microcracks within the sample, which can serve as defects for materials failure. Additionally, it is known that FFF-based printing process could result in layered 3D structures with limited interfacial adhesion. Here, we note that even though the samples were thermally annealed at 150 °C for 24 h during sulfonation, significantly anisotropic mechanical properties were still observed in their mechanical properties (Supplementary Fig. 13), suggesting the crosslinking step alone can not efficiently weld the interfaces. We attributed this result to several factors, including (1) commercial PP filaments are typically highly entangled and thus exhibit a slow diffusion rate, and (2) crosslinking of PP in acid occurs first at the interfaces between printed layers, which can kinetically trap polymer chains and further hinder their ability to perform inter-layer diffusion for effectively welding. Collectively, our carbon samples, derived from 3D printed PP-CF precursors, does not exhibit extraordinarily high mechanical stiffness and strength as their aerospace grade counterparts. However, we first time reported an approach to enable accurate manufacturing of structurally complex carbons, with very limited shrinkage from printed to carbonized state, as well as providing controlled mechanical properties with inherent elasticity and compressibility through simple varying processing time, providing significant advantages over prior arts, particularly for applications of wearable and deformable carbon-based electronic devices.

## Discussion

Enabling direct and accurate control over large-scale complex structures of carbon materials can broadly benefit the advancement of many technologies, including energy storage[54], catalysis[55], and environmental remediation applications[56]. In recent years, a variety of AM methods have focused on addressing this emerging need, including printing and pyrolysis of polymeric carbon precursors and/or solvent/matrix removal of carbon-containing inks; these previous efforts established a groundwork for creating structured carbons. However, most reported methods so far have several longstanding challenges that need to be addressed to enable AM of carbons at scale, including materials and instrument cost, process scalability, and accurate dimensional control over resulting parts after experiencing pyrolysis, hindering the scaled production and direct use of structured carbons in many important areas. Furthermore, previous methods for developing compressible and elastic carbons often involve complex processes and their integration with AM to on-demand producing structured parts is still in its infancy[51].

This work directly provides a transformative and industrially feasible approach to AM of carbons with controlled structure and mechanical properties, with following distinct advantages over prior arts: (1) using low-cost, widely available printing filaments as precursors, (2) materials exhibiting excellent dimensional stability upon conversion from printed to carbonized states, and (3) simple and scalable manufacturing process, which can be generalized to different fiber filler-containing PP filaments as precursors. Noteworthily, the <4%-dimensional shrinkage is significantly lower than most previous reports of preparing carbons from polymer precursors (Supplementary Table 2), allowing more streamlined manufacturing process from design to production with increased resource efficiency. This low shrinkage provides an opportunity to increase the manufacturing resource efficiency and product quality consistency, while the known low shrinkage degree can be accommodated in the initial step of product design. This work investigated a series of fiber-containing PP filament, including both CF and GF fillers, to confirm the generalizability of our process, showing a consistent advantage of very low volumetric shrinkage; the crack formation was all observed in these systems. We would like to note that as the crack formation is primarily

driven by the reaction-induced stress within printed PP, it could be possible to alter the crack initiation time and their averaged distance through varying the properties of PP resin (such as molecular weight and crystallinity) and/or inclusion of additives (such as plasticizers). However, these factors would not change the key observations and results of this work, including excellent structural retention enabled by fiber fillers and a crack-facilitated PP crosslinking mechanism. Moreover, our approach of converting structured PP to carbons through steps of sulfonation and pyrolysis can be extended to different AM methods to prepare precursor parts, including a PBF-based method. The change in precursor preparations may result in parts with different void formation and layer thickness, and thus may lead to altered crosslinking and cracking kinetics. On this end, further studies can focus on understanding and optimizing the key material and process parameters to control the diffusion of sulfuric acid into printed parts, while collectively considering factors of natural porosity, sample size, AM throughput, and system cost.

Furthermore, development of porous structures through under-crosslinking of precursors opens up great opportunities to attain distinct mechanical properties and density of final carbon parts using the same printed precursors. The ability to decouple volumetric shrinkage (determined by fiber loading content) and carbon yield (controlled by sulfonation/crosslinking time) for AM of carbon is particularly noteworthy, as it directly enables the control over carbon mechanical properties without sacrificing the degree of structural retention, addressing a previous significant barrier in the area known for the trade-off between shrinkage prevention and mechanical property. As an example, a recent work from Lu et al., demonstrated partial pyrolysis is required to enable lightweight and tough 3D printed carbons; increasing carbonization degree leads to stiffer samples with a larger degree of sample shrinkage occurred[11]. Collectively, we believe reported technology is highly adaptable and broadly applicable for making major contributions to the development of next generation 3D functional carbon materials.

We envision the lightweight feature and compressible and elastic mechanical behaviors of our 3D printed carbon, along with high specific strength-to-weight ratio can be useful for a variety of sensing applications including health monitoring as soft robotics, as well as environmental remediation through oil sorption and water purification and even as materials for energy generation and storage. Additionally, future work can optimize reaction condition and system design (e.g. filament composition and large-scale structures) to further improve the mechanical properties and process efficiency of these carbons, allowing their use for structural composites applications. As an example, Supplementary Figs. 25 and 26 show that by using fuming acid or increasing reaction temperature, faster PP-CF crosslinking kinetics can be obtained, resulting in significantly shorted reaction time (to ~2 h). Specifically, the presence of free $SO_3$ group in the fuming acid can improve PP reaction kinetics, while higher sulfonation temperature can also lead to enhanced reaction kinetics, accompanied with an earlier crack initiation time, and faster acid diffusion rate. Regarding with addressing the anisotropic mechanical properties, an inherent challenge from using FFF method, we note several approaches can be employed to weld interfaces between printed layers of PP-CF, prior to the sulfonation reaction, such as microwave heating[57], or development of core-shell filaments[58,59]. For these structured filaments, during thermal treatment shell can flow to increase isotropy while the core remains structurally in-tact. Moreover, highly efficient Joule heating performance of these materials coupled with tailorable geometries enable their high potential for the use in industrial decarbonization and electrification.

This work can also stimulate many exciting new research directions for future study on the topics of AM carbons, in addition to exploring the application domains. For example, many potential opportunities exist such as developing mechanical metamaterials with controlled porosity and mechanical responses, creating structurally graded materials through spatially selective crosslinking, extending fiber-reinforced polyolefin precursors to conventional polymer processing systems for carbon manufacturing with accurate structural control, and further strengthening material functionality and performance through the use of low-dimensional carbon additives.

## Methods

### Materials preparation

Braskem FL900PP-CF carbon fiber-reinforced polypropylene (PP-CF) filament with a diameter of 2.85 mm was purchased from Dynamism and used in this study. This filament contained ~15 wt% carbon fibers (CFs). CFs have an averaged diameter of 9.3 μm and length of ~300 μm. Additional carbon-reinforced PP was compounded from blending Braskem FL900PP and FLPP100PP to attain controlled CF content, which was then extruded through a Filabot single screw extruder and accompanying puller at a nozzle temperature of 220 °C and a screw speed of 15 rpm. The filament diameter was kept at 2.85 mm. FiberX PP-GF30 glass-filled filament containing 30 wt% glass fibers was purchased from Dynamism. 98 wt% sulfuric acid and fuming sulfuric acid (20% fuming, 18–24% free $SO_3$) were obtained from Fisher Scientific. Deionized (DI) water was obtained by passing tap water through a Milli-Q IQ 7003 ultrapure lab water purification system from Millipore Sigma.

3D printed specimens in this study were prepared using an Ultimaker S3 with an Ultimaker CC 0.6 mm print core. In a typical procedure, printing was performed at a nozzle temperature of 250 °C and a bed temperature of 80 °C under ambient conditions. A commercially available Magigoo adhesive was applied to the surface of print bed prior to FFF printing to prevent warp. Parts were printed with a printing speed of 40 mm/s and a fan speed of 20%, all parts were printed with a brim and without support material. Cubic gyroid structures pictured in Fig. 1a were printed using a gyroid infill pattern with a 20% infill density with no walls, top or bottom layers. The model system in Fig. 1b–e was a printed cubic gyroid structure (~15 mm in dimension) with an infill percentage of 20%. Cubic shrinkage specimens in Fig. 2a were printed with no walls, top, or bottom layers with a 40% infill density in a gyroid pattern. Samples for mechanical property test had cubic geometry 2 cm in dimension. Complex parts were printed with two walls (for objects listed in Fig. 3b), two top layers and zero bottom layers with an infill density of 20% and a gyroid infill pattern. STL files of all parts reported in this work can be found on Thingiverse.com.

In a typical crosslinking reaction, FFF printed PP-CF specimens were first fully submerged in concentrated sulfuric acid within beakers. They were then transferred to a Thermo Scientific Thermolyne F6010 muffle furnace and the temperature was increased by 2 °C/min until the sulfonation temperature (150 °C, unless otherwise noted) was reached. For crosslinking of PP-CF, isothermal conditions were maintained for a controlled amount of time. Specimens were then removed from their glass containers and washed three times with DI water to completely remove residual acid and other reaction by-products. The neutralization of acid waste was confirmed using pH paper. Samples were then rinsed with acetone to facilitate drying and placed under vacuum in a vacuum oven overnight. For reaction safety, proper personal protective equipment must be worn to mitigate any risks as concentrated sulfuric acid can cause harm to users upon exposure.

Completely dried sulfonated samples were placed in an Across international TF1400 tube furnace tube furnace under an inert nitrogen environment for carbonization. A ramp rate of 1 °C/min was used from ambient temperature up to 600 °C after which a 5 °C/min rate was used until the desired carbonization temperature (e.g. 800 °C) was reached. After carbonization, the furnace was allowed to cool naturally to ambient temperature, which took ~4 h (from 800 °C to 25 °C).

## General characterization

A TA Instruments Discovery 550 TGA was used to determine filler content in both Braskem FL900PP-CF and as-compounded PP-CF filaments using a 10 °C/min ramp from room temperature to 800 °C under nitrogen. Bulk carbon yield for the model system was determined using an analytical balance to measure the as printed PP-CF parts, which was compared to the resulting mass of the corresponding final carbon. Mass increase during the sulfonation process was similarly measured by comparing the mass of completely dried sulfonated samples to their initial mass before treatment. Differential scanning calorimetry (DSC) was performed using a TA Instruments Discovery 250 DSC with T-Zero pans. A heat-cool-heat cycle was employed with an initial heating cycle from 20 °C to 220 °C at a rate of 10 °C/min to erase thermal history. Samples were cooled to 20 °C at a rate of 5 °C/min and then heated back to 220 °C at a rate of 10 °C/min. Degree of PP crystallinity in these samples was determined through integration of the melting peak of the second heating cycle, adjusting for the presence of fiber fillers within the system and comparison the result to the theoretical enthalpy of melt of 100% crystalline PP (-207 J/g). Data analysis of DSC and TGA experiments was performed using Trios software. PP melting temperature is found to be -155 °C. A PerkinElmer Frontier attenuated total reflection FTIR spectrometer was used to monitor changes in the chemical composition of sulfonated parts as a function of time. The scan range was 4000–600 cm$^{-1}$ with 32 scans and a resolution of 4 cm$^{-1}$. Data analysis was performed using Wavemetrics Igor Pro 9.

A Zeiss Ultra 60 field-emission SEM was used to understand morphological changes in the sulfonation parts after various crosslinking time, as well as after the carbonization process, with an accelerating voltage of 10 kV. Prior to SEM imaging, parts were carbon coated using a 208 Cressington Carbon Coater. Crack-to-crack distance was assessed using Image J analysis software. Dimensional shrinkage between the initial prints and the final carbon structures was assessed through the measurement of critical dimensions (including length, width, and height) of as printed and carbonized samples. The averaged shrinkage value was determined over five samples. Solid framework density of PP-CF derived carbons was measured using an Anton Paar Ultrapyc 5000 pycnometer under helium for 15 replicate cycles and an analytical balance. Apparent density of the printed carbons was assessed by measuring the printed parts using calipers, calculating the volume occupied by the part and dividing by the mass as assessed by use of an analytical balance. Nitrogen physisorption experiments were conducted using a Tristar II 3020 (Micromeritics) and Brunaur-Emmett-Teller (BET) surface area and Barrett-Joyner-Halenda (BJH) pore size distribution analysis. Porosity of the PP-CF derived carbon was determined based on the density of the carbon framework measured by pycnometery, the calculated framework density of the PP-CF derived carbon, and the known density of air, using following equation: $\varnothing = \left(1 - \frac{\rho_{bulk}}{\rho_{particle}}\right) \cdot 100\%$, where $\varnothing$ is the porosity expressed as a percentage, $\rho_{bulk}$ is the bulk density of the printed structure and $\rho_{particle}$ is the density of the fully dense particle as determined by helium pycnometry. Raman spectroscopy was performed using an Andor Kymera 328i spectrometer which was equipped with an Andor Newton camera. For these experiments the laser was operated at 532 nm and with a 20 mW power.

Compressive mechanical testing was performed in accordance with a modified ASTM D695 standard using an MTS (i.e. Material Testing System) Insight test frame with a 5 kN load cell and compression grips; a strain rate of 1 mm/min was used. Mechanical property data was analyzed using Igor Pro 8 to identify compressive yield strength by the point of zero slope in the stress strain curve and compressive modulus through the initial slope of the linear elastic regime. Similarly, the effect of fiber inclusion on tensile properties of PP was determined through tensile testing of ASTM D638 type 1 tensile bars, tested in tension on an MTS insight test frame with a strain rate of 5 mm/min. Cyclic compressive testing was performed using a Mark-10 F105 test frame with compression grips. For these experiments a strain limited compressive test was performed with a strain rate of 50 mm/min and a determined end point of 20% or 40%. The Joule heating performance of PP-CF-derived carbon was determined by connecting the carbonized parts to a DC power supply (from Dr. Meter) using ceramic tile as a support. The power was increased incrementally, and the temperature was measured using a Proster K-type thermocouple.

## Computational models and methods

The multiscale computation consists of models at three length scales: (1) the 3D-printed macroscopic gyroid structure, (2) the mesoscopic composite representative volume element (RVE) with random fibers and pores embedded in the matrix, and (3) the polymer-derived carbon matrix. Finite element modeling was conducted at both macroscopic and mesoscopic levels. The mesoscopic analysis, particularly, assesses the effective mechanical properties of the PP-CF derived carbon composites, which are used in the macroscopic analysis as the model material. At the macroscale, geometry of the cubic gyroid structure was created based on the desired dimensions and in-fill density. The model was meshed with 40,537 nodes and 22,347 ten-node tetrahedral elements. A displacement-controlled loading was imposed on the top surface of the gyroid structure, while the bottom surface was restrained.

At the mesoscale, several RVEs with the same fiber fraction but different void volume fractions were developed to represent the PP-CF-derived carbon composites treated with different sulfonation times. A macro-porosity of 0% corresponds to a sulfonation time of 24 h, signifying the sample from a completely crosslinked PP matrix. In contrast, sulfonation times of 18 h, 12 h, 8 h, and 6 h correspond to macropore porosities of 1.9%, 5.8%, 28.4%, and 49.6%, respectively, indicating that the 3D printed carbon derived from PP matrix with a reduced degree of crosslinking. While voids in realistic composites may possess complex shapes, this study assumed circular voids with the same radius as the fibers, which reasonably agrees with experimental characterizations. The edge length of RVEs' model is 114.31 μm, which refers to a diameter of 10 μm for carbon fibers and embedded voids. To further investigate the effects of void shape, RVEs with elliptical voids and randomized pore sizes were generated with eccentricities ranging from 0 to 0.9 and random major axis orientations (Supplementary Figs. 27 and 28). As shown in Supplementary Figs. 29 and 30, void shape, anisotropy factor, and size distribution all had very limited impact on changing the mechanical properties of PP-CF derived carbon materials, justifying the assumption of circular voids. To rigorously apply micromechanics, all RVEs were generated to possess geometric periodicity. When a fiber or void crossed the RVE boundary, the portion not included in the RVE was repositioned to the opposite side to ensure periodicity. The meshing of the RVEs was specially conducted to retain periodicity, ensuring that any two opposite faces possessed the same surface mesh. The periodic mesh enabled the implementation of periodic boundary conditions, which was essential to preserve continuity in displacement and stress fields across RVE boundaries. Global strains were applied to RVEs to induce stresses where were averaged for the calculation of global stresses. The global stress-strain behavior of the carbons RVE led to the deduction of key material properties of the composite which was then applied to the global cubic gyroid model.

In terms of material models, carbon fibers were considered transversely isotropic, linear elastic material that is not damageable due to the inherent high strengths. Key elastic properties include the longitudinal and transverse elastic moduli of the fiber, $E_1$, and $E_2$, respectively. Additionally, $\nu_{12}$ denotes the Poisson's Ratio of the fiber, while $G_{12}$ and $G_{23}$ correspond to the longitudinal and transverse elastic shear moduli of the fiber. It is noteworthy that the CF properties remain unchanged with different sulfonation times.

The PP-derived carbon matrix was modeled as an elastoplastic and potentially damageable material. Herein, $E$ stands as the elastic modulus of the matrix in all directions, and $\nu$ signifies the Poisson's ratio of the matrix. A damage-plasticity model based on the Drucker-Prager yield surface[60], with modifications proposed by Lee and Fenves, was utilized[61]. This model allows distinct tensile and compressive yield points for the material. The damage initiation process in the PP matrix is considered to commence with a minimal plastic strain that can be deemed negligible, thereby equating the tensile yield stress to the tensile strength, $UTS_t$. Furthermore, $\sigma_{yc}$ represents the compressive yield stress. Moreover, the model incorporated a single variable, fracture energy $G$, to control energy dissipation after damage initiation in tension. The interface damage between neighboring fiber and matrix elements was modeled using cohesive elements automatically inserted at the interface. The initiation of damage within the interface cohesive elements was ascertained using a quadratic traction-interaction criterion that encompasses the nominal stress ratios along the three principal directions signified by the subscripts: the normal direction, $n$, the first shear direction, $s$, and second shear direction, $t$. $t^0$ symbolizes the peak values of the nominal stress during deformation[62]. Damage evolution was governed by a power-law mixed-mode fracture criterion. $G^c$ is the critical energy release rate. The properties of the PP matrix displayed variations due to distinct synthesis and processing conditions. To understand how varying sulfonation times impact the mechanical properties of the PP-derived carbon matrix, material properties for both the carbon fiber and matrix are tabulated in Supplementary Table 3.

## Data availability

The authors declare that data supporting the findings of this study are available within the paper and its Supplementary Information Files. Data generated in this study are provided in the Source Data file with this paper. Data are also available from the corresponding author upon request. Source data are provided with this paper.

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

## Acknowledgements

This project was partially supported by the start-up funding from the University of Southern Mississippi, Mississippi SMART Business Act, and National Science Foundation (No. CMMI-2239408). The computational work was supported by Temple University' startup grant, the U.S. Department of Energy Advanced Research Projects Agency–Energy (ARPA-E) (DE-AR0001576), and the Advanced Materials and Manu-facturing Technologies Office (AMMTO) (DE-EE0010205). The authors thank the help from Dr. Shan Yang for assisting the Raman measure-ments. The authors are also grateful to Drs. Jeffery Wiggins, Derek Patton, and James Rawlins for access to their equipment and useful discussions.

## Author contributions

Z.Q. and P.S. designed experiments for this work, where P.S. conducted most of them. J.H. and L.L. designed and performed computational work. M.R., A.G.O., A.G. and E.B. assisted in data collection throughout the work. P.S. and Z.Q wrote the manuscript, which C.B.D. and C.Y. also contributed to. L.L and Z.Q. supervised the entire project.

## Competing interests

The authors declare the following competing interests: Z.Q., M.R. and P.S. submitted two U.S. patent applications for relevant technology of AM of carbons (U.S. Application No. 17/848,342; No. 18/112,446). The remaining authors declare no competing interests.
