## [Peer Review File · Nature Communications]

Accurate additive manufacturing of lightweight and elastic carbons using plastic precursorsREVIEWER COMMENTS

Reviewer #1 (Remarks to the Author):

This research nicely takes the earlier work in an exciting new direction with the inclusion of carbon fiber reinforcement with the existing transformation of waste polypropylene to CF reinforced carbon. The manuscript represents a comprehensive treatment of the chemical transformation and demonstrates fundamental understanding of PP carbonization together with CF. The printing and characterization clearly portray the discovery in a very enabling manner using extrusion-based AM (FFF).

A more thorough discussion of the role of carbon fiber surface and its effect on mechanical performance would be an important question to address in more detail. Although the work is clearly novel and builds on earlier work, the article might be considered either in an additive manufacturing-oriented journal or a composites-based journal. In addition, *Advanced Materials* would also be ideal to capture the largest number of citations with relevant readership.

Overall, the manuscript is thorough, well written, represents an excellent next step in their research. Moreover, the manuscript represents an exciting intersection of polymers, AM, and sustainability.

Reviewer #2 (Remarks to the Author):

Smith et al. reported 3D printed elastic carbon materials using commercially available polypropylene (PP)/carbon fiber (CF) composite with low dimensional shrinkage (< 4%). The use of polyolefin as a carbon source is an emerging need to address the sustainability challenges of plastic waste. In addition, precise dimension control is essential to harness the full capability of additive manufacturing in general. To this end, this work successfully demonstrated the fabrication of elastic carbon materials with tunable porosity by changing the sulfonation reaction time. Overall, the manuscript is well-written and carefully studied. However, there are some points to be addressed as detailed as follows:

Major comments:

1. A seemingly unavoidable trade-off between shrinkage prevention and mechanical property: As the authors described in the last paragraph, the reported approach sacrifices the mechanical property to use commercially available PP-CF and minimize the volume shrinkage. Because preventing volume shrinkage inherently relies on the micropore formation in the structure, this is extremely challenging to overcome. While this manuscript is well-written and carefully investigated the system, it might have a limited impact in the field as well as room for improvement. The author's perspective on this point would be appreciated.
2. Homogeneity of the sample: Considering that the degree of carbonization, which in turn dictates the porosity, is controlled by sulfonation reaction, one might expect structural heterogeneity from surface to inside. While the dimension control is successful across different sizes (Figure 3a), is there any density gradient in the structure?
3. Chemical identity of PP-CF: The commercially available PP-CF resin used in this study should have some additives to improve the dispersion of CF and adhesion between PP and CF. How do such additives influence crack formation? Does just a simple blend of PP-CF yield the same result? This would be crucial insight to generalize this approach more broadly to olefin/CF resin systems that may perform better.
4. Risk assessment on sulfonation reaction: The long sulfonation reaction with concentrated sulfuric acid at 150 C is a high safety-risk experiment. I recommend leaving some comments about safety. Also, seeking alternative reaction pathways would be very impactful to make this process more accessible.
5. Joule heating: While the heating property in Figure 4 is impressive, a proper comparison with benchmarks from previous literature would be helpful.
6. RVE analysis is a bit confusing because the fiber is not clearly illustrated in Figure 7b and S22,

and there is no clear description of the length scale (the voids are assumed to be the same diameter as CF, so $\sim 9 \mu\text{m}$?). It was also surprising that the isotropic assumption seems to work well enough even though the structure is highly anisotropic due to the FDM process.

Minor comments

1. Page 8, line 156: Easier to follow if Figure S5a is pointed out as a schematic for the reaction.
2. Figure 7e: Error bars should be included.

Reviewer #3 (Remarks to the Author):

This is a reviewer response to the manuscript titled "Lightweight and elastic carbons from 3D-printed plastics: accurate structural control enabling tunable mechanical properties." The title is well-suited to the article as it describes a process of transforming polypropylene objects fabricated through fused filament fabrication (FFF) additive manufacturing (AM) into carbon. The titular "structural control" arises from varying the time allowed for sulfonation reaction (a post-print submersion in sulfuric acid). The authors demonstrate that removal from the sulfuric acid bath prior to what they term "complete sulfonation" yields unsaturated double bonds and uncrosslinked polypropylene that reduces the density of the final, carbonized structure through liberation of these moieties in the post-acid bath carbonizing process (high temperature pyrolysis in an inert atmosphere). The authors show that removing the parts from the acid bath at different times only affects the porosity of the structure without compromising their carbon yield. Therefore, compression properties can be tuned while maintaining the unique properties of carbon structures, including Joule heating, which the authors demonstrate.

This manuscript extends the authors' previously published work (Smith, et al. 2023; Reference [24]) through the inclusion of chopped carbon fiber (CF) in the PP filament. The evident difference in resulting carbon structure performance is noteworthy. The authors describe how including the carbon fibers aids in holding the printed structure together during the sulfonation, drying, and carbonization process. The increased stiffness and imbalance of sulfuric acid uptake results in oriented microcracks as the PP swells and CF does not. Although these cracks can be argued to diminish overall mechanical performance, the authors point out that without these cracks, full sulfonation (and therefore crosslinking) cannot occur since this is a diffusion limited process. Additionally, including CF significantly reduces shrinkage throughout the post-print chemical modification and conversion into carbon; this is a significant improvement over other published methods describing conversion of AM parts into carbon.

The present work is highly impactful to a broad audience as the demonstrated means of converting an AM part to a carbon structure is accessible. The authors use commercially available off-the-shelf FFF machines (Ultimaker) and PP filament (Braskem), sulfuric acid is a common chemical, and the high-temperature furnace for carbonization is also commercially available, albeit likely the most expensive aspect of the set-up. Importantly, I believe I could replicate the results in my lab based on the information provided in the report. There is clearly much future work to be done in the characterization and application development for the described method – far more than a single research group can accomplish, which is why I recommend publication of this manuscript in Nature Communication.

While the submitted work is impactful and sufficient for publication (following the minor revisions suggested), it has left the Reviewer with a few thoughts and questions. Given the broad readership of Nature Communication, it is reasonable for these items to have been omitted from the submitted manuscript, but should certainly be addressed for a more complete depiction of the described process. The Reviewer notes the following topics of further discussion: (i) comparison of diffusion against other AM carbonization processes, (ii) the inseparable impact of part design on the carbonization process, and (iii) greater depth on the micro/mesoscale influence of the CF on carbonization. Naturally, the concerns of diffusion and design are linked and will be addressed together.

Other publications describing carbonization of AM parts from polyimides involve an outgassing process inherent to the imidization reaction (e.g., Arrington, et al. 2021; Reference [12]). This process often results in catastrophic failure as the expanding gas generated from inside the bulk of the part creates large cracks as it escapes, similar to the combined burnout-sintering common to

binder jetting (Rahman, Wei, Miyanaji, and Williams Addit. Manuf. 2023) and fabricating ceramic structures through vat photopolymerization of highly loaded polymer resins (Cao, et al. Addit. Manuf. 2021). It seems that the process described in the present manuscript has exchanged the challenge of inside-to-outside diffusion of a gas for the outside-to-inside diffusion of the sulfuric acid. The authors state in the present work that the microcracks that form through the swelling strain mismatch between the CF and the PP matrix aid in deep penetration of sulfuric acid, and therefore thorough crosslinking throughout the printed PP structure. Additional work should understand the nature of the crosslinked network (e.g., molecular weight between crosslinks) and its dependence on sulfuric acid concentration and diffusion rate. All parts shown in the present manuscript are reported to have been printed at 20-40% infill; this allows for an ample surface area to volume ratio that limits the maximum distance into the part required for sulfuric acid penetration. The Reviewer wonders whether the described process could successfully create solid structures of carbon at any length scale, or what the maximum feature size might be. Fortunately, AM techniques excel at lattice-type structures so the issue of penetration depth may be mitigated through intelligent, science-based design; however, the Reviewer is not aware of the existence of any design guidelines for such an effort.

In addition to "design for diffusion," the effects of overall design choice cascade into carbonization and final mechanical properties. Although tailorable from a bulk sense, the mechanical properties of the final carbonized part have not been shown to be functionally graded, which is a desired ability in AM fabrication. Selective carbonization would be helpful for realizing so-called "smart structures" that combine structure and sensing capabilities. This does not seem possible with the currently presented process.

Finally, the Reviewer would like to know more regarding the micro and mesoscale influence of the CF on the carbonization process. The current manuscript focuses on the stiffness of the already-carbonized CF and how that benefits the carbonization of the matrix PP throughout acid swell and pyrolysis steps. It is unclear from the present manuscript if there is a templating effect of the CF either on the sulfonated PP or the final carbonized structure. Regardless of templating, it would be especially interesting to discover whether the alignment of carbons in the fiber regions is preserved following pyrolysis. If so, design (including toolpath planning) again becomes a critical feature.

The Reviewer is admittedly unfamiliar with the field of carbonization; however, I estimate that carbons derived from PP should have a maximum yield of 86 %wt in relation to the starting mass of PP based on the weight ratio of carbon to hydrogen in the repeating unit. The authors state achieving 61% yield in the present work and provide two citations (one being the authors' prior work; Reference [50]) to justify this value as typical. The Reviewer is curious where the balance of the mass goes; is it vaporized as carbon dioxide through a reaction with the sulfuric acid? Neither the current work nor the provided citations offer an explanation. Including an additional reference that covers this rationale would enhance the value of the manuscript to a broader audience.

The Reviewer requests for the authors to include some discussion of annealing and crystallization as part of the discussion of sulfonation. The authors describe performing sulfonation at 150 °C, which although is below the observed peak melting temperature of PP (according to Figure S3), is certainly above its glass transition temperature. The authors state that a common issue in FFF parts is anisotropy due to layering for both strength and shrinkage. However, if held above glass transition temperature for multiple hours, one would expect some degree of macro-scale healing at the layer interface to alleviate this issue and promote increase isotropy. Often, annealing is impractical for FFF parts as it would induce slumping at these elevated temperatures. The Army Research Labs has reported on successful annealing of FFF printed objects using a multi-material core/shell approach where the core remains structurally in-tact at the chosen annealing temperature while the lower temperature shell can reflow enough to increase isotropy and strengthen weld lines (Toal, Holmes, Rodriguez, and Wetzel Addit. Manuf. 2017). It is reasonable that a combination of the CF, the PP crystalline fraction, and the on-going crosslinking is able to provide the support at 150 °C to prevent slumping or macro-scale part distortion. It would be interesting to know the relation between crosslinking kinetics (i.e., rate) and rate/degree of healing of the weld interfaces as a function of temperature. Perhaps isotropy could be increased if the process began at a lower temperature for a longer time?

In the Discussion section, the authors make statements that appear counter to what has been argued earlier in the manuscript. For instance, the Discussion section identifies "instrument cost, process scalability, and accurate dimensional control" as "intractable" challenges. The Reviewer estimates the printer and filament used in this work to total < \$5,000. The authors themselves

identify a key advantage of their process to be “low cost” (page 26). Which aspect of the process is cost prohibitive? On page 5, the authors report that the presented process is “simple and scalable.” Please explain how that can be true on page 5 but “scalability” is an intractable challenge on page 25. Finally, please clarify what would be an appropriate target tolerance for the intractable challenge of dimensional control. Especially please clarify whether this is a challenge unique to the presented carbonization process or else a more general challenge for the FFF modality of AM (which, for example, could be improved simply through using a more expensive, higher precision machine like a Stratasys instead of an Ultimaker).

Minor Revisions

Given the broad readership of Nature Communications, it is understandable that the manuscript contains what are frequently used terms by the general public to refer to different AM technologies. However, the Review strongly encourages the authors to revise the manuscript to include the standardized ASTM/ISO 52900 terms for AM technologies. This would include swapping the trademarked term “fused deposition modelling (FDM)” for the generic “fused filament fabrication (FFF)” or “filament material extrusion.” Additionally the manuscript references the legacy trademarked term “selective laser sintering,” which should be revised as “laser bed powder bed fusion (PBF).”

In a similar vein, please reserve “filament” for the feedstock form of the material and use the terms “layers” and “roads” to refer to material extruded during the printing process.

Please be sure that all references are correctly and consistently formatted in the Reference section. There is currently a mix of full author lists alongside “et al.” entries. Additionally, certain entries have inappropriate usage of italics or else are missing an appropriate usage.

On page 14, the authors cite Reference [57] at the end of a sentence describing work published by “Liao, et al.” However, Reference [57] is not this work. Please ensure correct in-text citations for this and all other instances.

On page 15, there is a sentence with the phrase “...to prepare a variety of carbon materials containing complex geometries...” Is it not more accurate to state that the authors have prepared “a variety of complex geometries containing carbon materials?” As the Reviewer understands the work, the carbon produced does not vary (e.g., in D/G ratio) from part to part; only the porosity (i.e., a component of part geometry) is affected.

On page 24, the authors use both the word “micropores” and “nanovoids” apparently referring to the same physical phenomenon. Please clarify which length scale most accurately represents these pore structures and revise accordingly.

RESPONSES TO REVIEWER COMMENTS

Reviewer #1 (Remarks to the Author):

This research nicely takes the earlier work in an exciting new direction with the inclusion of carbon fiber reinforcement with the existing transformation of waste polypropylene to CF reinforced carbon. The manuscript represents a comprehensive treatment of the chemical transformation and demonstrates fundamental understanding of PP carbonization together with CF. The printing and characterization clearly portray the discovery in a very enabling manner using extrusion-based AM (FFF).

Response: We thank the reviewer for carefully reviewing our work and providing constructive feedback. We have addressed the comment in our revised manuscript and the response is provided below.

A more thorough discussion of the role of carbon fiber surface and its effect on mechanical performance would be an important question to address in more detail.

Response: We agreed the role of carbon fiber surface can influence the interfacial interactions between carbon fiber and polymer-derived carbon matrix, potentially impacting the mechanical properties of their derived composites.

To address this comment, we first note that in our process, sulfuric acid was employed to crosslink PP-CF (commercial polypropylene filaments containing carbon fibers), which can react with both PP matrix and CF surfaces. The sulfonation of both components may enable their favorable interactions, which could be beneficial to further stabilize the interfaces. It is important to emphasize that sulfonation is a strong reaction; as a result, the pre-treatment of CF before their introduction into PP matrix prior to printing may have a very limited impact as the surface groups on the CF will be further oxidized during the crosslinking.

In our work, significantly different mechanical properties of 3D printed PP-CF derived carbons can be obtained under an identical printing-crosslinking-carbonization process, except only one variable was used (i.e. crosslinking time). This is due to the introduction of macropores, resulted from the minimal sample volumetric shrinkage upon carbonization even carbon yield is reduced. Simulation results in Figure 7 further confirmed this mechanism.

In the revised manuscript, we have now performed additional simulation work to understand the impact of CF-carbon matrix interfacial interactions on the mechanical properties, which could inform and motivate some future work to innovatively engineer the surface groups of CF after sulfonation process. As shown below, we selected the RVE (representative volume element) of the carbon prepared by 24 h crosslinking time as the representative. The material properties of the CF-carbon matrix interface are varied in the range from 50% (weakened) to 150% (strengthened). The left panel shows the compressive stress-strain curves, while the right panel shows the tensile stress-strain curves. The strength of the CF-carbon matrix interface is shown to

have very limited impact on the compressive mechanical behaviors; however, with stronger interfacial interactions, the yield point of composite under tension is delayed, and a higher strength is achieved. These results confirm the impact of carbon-fiber/carbon matrix interactions on controlling the overall mechanical properties of derived composites. We have now included these results and discussions in the revised manuscript.

Figure caption: Stress-strain curves of the RVE microstructure with varied interfacial strength for samples with 24 h crosslinking time, including compression and tension results.

“Furthermore, simulation was performed to understand the impact of fiber/matrix interfacial strength on the mechanical properties of derived 3D printed carbon composites; this is for informing future technology improvement to allow design of products with enhanced mechanical performance. Although we recognized that pre-treatment of fiber surface prior to filament preparation, printing, and crosslinking may have a limited impact as the sulfonation reaction may lead to altered functionality, opportunities may exist to engineer fiber surface after crosslinking. In this work, we selected the RVE of sample that was crosslinked for 24 h as a representative model for investigation. The material properties of the CF-carbon matrix interface were varied in the range from 50% (weakened) to 150% (strengthened). As shown in Figure S24, it was found that the strength of the CF-carbon matrix interface has very limited impact on the compressive mechanical behaviors; however, with stronger interfacial interactions, the yield point of composite under tension is delayed, and a higher strength can be obtained.”

Although the work is clearly novel and builds on earlier work, the article might be considered either in an additive manufacturing-oriented journal or a composites-based journal. In addition, *Advanced Materials* would also be ideal to capture the largest number of citations with relevant readership.

Response: We thank reviewer for this suggestion. We believe *Nature Communication* is a very suitable platform to disseminate our work due to its wide readership from various areas, including chemistry, polymers, additive manufacturing, biomaterials, and mechanical engineering; our work can provide contributions to researchers from all these areas. Additionally, a key advantage of our work is using a simple process and commercially available materials, which may be interesting for broader communities, including industry, to adapt our approach for accurate

additive manufacturing of carbons. On this end, the benefits of open access of Nature Communication are clear and strong.

Overall, the manuscript is thorough, well written, represents an excellent next step in their research. Moreover, the manuscript represents an exciting intersection of polymers, AM, and sustainability.

Response: We thank again for this very positive feedback!

Reviewer #2 (Remarks to the Author):

Smith et al. reported 3D printed elastic carbon materials using commercially available polypropylene (PP)/carbon fiber (CF) composite with low dimensional shrinkage (< 4%). The use of polyolefin as a carbon source is an emerging need to address the sustainability challenges of plastic waste. In addition, precise dimension control is essential to harness the full capability of additive manufacturing in general. To this end, this work successfully demonstrated the fabrication of elastic carbon materials with tunable porosity by changing the sulfonation reaction time. Overall, the manuscript is well-written and carefully studied.

Response: We thank the reviewer for carefully reviewing our work and providing very valuable feedback. The reviewer's very positive feedback is appreciated. We have addressed all comments to improve the quality of work, which are shown below.

However, there are some points to be addressed as detailed as follows:

Major comments:

1. A seemingly unavoidable trade-off between shrinkage prevention and mechanical property: As the authors described in the last paragraph, the reported approach sacrifices the mechanical property to use commercially available PP-CF and minimize the volume shrinkage. Because preventing volume shrinkage inherently relies on the micropore formation in the structure, this is extremely challenging to overcome. While this manuscript is well-written and carefully investigated the system, it might have a limited impact in the field as well as room for improvement. The author's perspective on this point would be appreciated.

Response: We thank the reviewer for this opportunity to clarify the novelty and impact of our work.

First, our work does not have a trade-off between shrinkage prevention and mechanical property, which is a key novelty and can make an important impact to the field. We note that in this work, the degree of shrinkage (from printed to carbonized systems) in PP-CF systems is only determined by the fiber loading content. As shown in figures below, for PP-CF with 15% fiber loading content, the dimensional shrinkage of both in-plane and out-of-plane directions for samples with different

crosslinking time (which leads to altered carbon yield) are consistently low, which is approximately 2% and 4%, respectively. In addition, we found that by reducing the fiber loading content, the dimensional shrinkage increased. These results further confirm the role of fiber fillers to restrict volumetric contraction of polymer matrix during their conversion to carbonaceous products, which is similar to several reports with pyrolysis of high filler-loading content systems (e.g. *Carbon*, **2017**, 115, 629; *Adv Mater*, **2023**, 35(38), 2208230). We also confirmed that our low minimal shrinkage in this process is preserved even increasing the carbonization temperature to 1400 °C (in SI), and can be extended to PP with glass fiber filler systems (in SI).

Figure caption: In-plane shrinkage of PP-CF derived carbons as a function of their crosslinking time

Figure caption: Dimensional shrinkage from converting printed PP-CF to their carbonized parts upon the inclusion of fiber at different loading levels between 0 and 10 %.

Because of the unique ability to decouple volumetric shrinkage (determine by fiber loading content) and carbon yield (controlled by sulfonation/crosslinking time), our work provides an exciting advantage to overcome a previous significant barrier in this area (just as the reviewer mentioned). A great example of prior art is a recent piece of elegant work from Yang et al (*Matter*, **2022**, 5, 4029), demonstrating that to enable lightweight and tough 3D printed carbons, optimization of carbonization degree is required for controlling partial pyrolysis, which leads to a trade-off between shrinkage and mechanical property. Here, our system can consistently yield very low shrinkage from printed to carbonized parts, while providing the ability to alter mechanical properties. We have now revised our manuscript to clarify this point, as shown below:

“The ability to decouple volumetric shrinkage (determined by fiber loading content) and carbon yield (controlled by sulfonation/crosslinking time) for AM of carbon is particularly noteworthy, as it directly enables the control over carbon mechanical properties without sacrificing the degree of structural retention, addressing a previous significant barrier in the area known for the trade-off between shrinkage prevention and mechanical property. As an example, a recent work from Yang et al., demonstrated partial pyrolysis is required to enable lightweight and tough 3D printed carbons; increasing carbonization degree leads to stiffer samples with a larger degree of sample shrinkage occurred.”

“Additionally, future work can optimize reaction condition and system design (e.g. filament composition and large-scale structures) to further improve the mechanical properties of these carbons, allowing their use for structural composites applications.”

2. Homogeneity of the sample: Considering that the degree of carbonization, which in turn dictates the porosity, is controlled by sulfonation reaction, one might expect structural heterogeneity from surface to inside. While the dimension control is successful across different sizes (Figure 3a), is there any density gradient in the structure?

Response: We appreciate this question and agree that understanding the sample heterogeneity is important. In our system, while changing degree of sulfonation/crosslinking leads to varied pore formation within our samples, the local morphology and properties of carbon are still consistent in different areas. This is due to the low in-fill density (~20%) of printed parts to allow acid to quickly diffuse within the samples, while FFF printing process leaves inherent void spaces between printed layers as shown below. This also explains why our dimensional control can be successful across different sizes, using an identical crosslinking procedure.

Figure caption: Characteristic void features of FFF printed parts leading to facilitated acid diffusion among different printed layers

To further confirm that there is no density gradient in the structure, we measured the pore textures and characterized SEM images at different locations of a piece of carbon sample (~3 cm in length). This sample was crosslinked for 12 h. As shown in below, the SEM images show similar

cracking behaviors occurred in these PP-CF derived carbon samples, while their surface areas (in the range of 360-380 m²/g) and porosity (63-67%) are also consistent.

Figure caption: Local SEM image and surface at different locations of a large printed structure

3. Chemical identity of PP-CF: The commercially available PP-CF resin used in this study should have some additives to improve the dispersion of CF and adhesion between PP and CF. How do such additives influence crack formation? Does just a simple blend of PP-CF yield the same result? This would be crucial insight to generalize this approach more broadly to olefin/CF resin systems that may perform better.

Response: We appreciate this question. While the model system in this work was focused on a commercial filament of PP-CF from Braskem (Braskem FL900PP-CF), we also investigated another system of glass fiber containing PP, which the same effect of very low degree of shrinkage was observed. We agree that these commercial filaments typically contains a very low amount of additives for improving processability, but they will only have a very minimal (at most) impact on the cracking formation mechanism, as the cracking mechanism is due to the mismatched volumetric change between outside and inside layers; during sulfonation reaction, most additives are susceptible to degradation as well; in a previous work, we found the cracks of PP were still formed with the absence of CF in the filament (*Adv Mater*, **2023**, 35(17), 2208029). The result is shown below.

Editorial Note: Panels below reproduced from Smith, P. et al. Additive manufacturing of carbon using commodity polypropylene. *Adv. Mater.* **35**, 220829 (2023), with permission from John Wiley and Sons. <https://doi.org/10.1002/adma.202208029>

Figure caption: SEM images of crosslinked PP samples (no fiber fillers) as a function of reaction time at 150 °C. Scale bar: 300 μm.

Moreover, we performed additional experiments to test our hypothesis that additives will not impact crack formation mechanisms and resulting low degree of shrinkage. We physically blended the same CF (recovered from the filaments (Braskem FL900PP-CF) through pyrolysis at 800 °C, which all organics are degraded) with another commercial PP (extrusion-grade) at 15 wt% loading content, and used our method to produce a 3D printed carbon. It is found that the same cracking behavior was occurred, while in-plane and out-of-plane shrinkage was 5% and 2%, respectively. These results are very similar to the model commercial filament system (Braskem FL900PP-CF). While we understand that with introduction of additive may alter polymer properties, which can influence very detailed specifics of cracking, such as crack-to-crack distance and a slight difference in crack initiation time, it is also clear the additives themselves will not change the key observation and results of our work. Following clarifications are provided in the revised manuscript:

“This work investigated a series of fiber-containing PP filament, including both CF and GF fillers, to confirm the generalizability of our process, showing a consistent advantage of very low volumetric shrinkage; the crack formation was all observed in these systems. We would like to note that as the crack formation is primarily driven by the reaction-induced stress within printed PP, it could be possible to alter the crack initiation time and their averaged distance through varying the properties of PP resin (such as molecular weight and crystallinity) and/or inclusion of additives (such as plasticizers). However, these factors would not change the key observations and results of this work, including excellent structural retention enabled by fiber fillers and a crack-facilitated PP crosslinking mechanism.”

4. Risk assessment on sulfonation reaction: The long sulfonation reaction with concentrated sulfuric acid at 150 C is a high safety-risk experiment. I recommend leaving some comments about safety. Also, seeking alternative reaction pathways would be very impactful to make this process more accessible.

Response: We agreed with this comment. We have now included comments in both Experimental and Results sections to increase cautious, and reflect the potential safety concern. Specifically, a highly relevant paper is now cited (*Ind. Eng. Chem. Res.* 2018, 57, 18, 6123–6130) to provide readers the information. This work was performed by the Dow Chemical Company, who investigated scaled production of carbon fibers using polyethylene, in conjugation with a similar sulfonation-induced crosslinking method.

“the use of sulfuric acid requires extra cautious for reaction safety and waste management, which the relevant information can be found within a work demonstrating a continuous multiphase reactor for polyolefin sulfonation and crosslinking toward carbon fiber production.”

“For reaction safety, proper personal protective equipment must be worn to mitigate any risks as concentrated sulfuric acid can cause harm to users upon exposure.”

5. Joule heating: While the heating property in Figure 4 is impressive, a proper comparison with benchmarks from previous literature would be helpful.

Response: We agreed that the Joule heating performance of our 3D printed carbon is excellent. In general, according to Joule’s law, the Joule heating temperature is directly controlled by the sample’s electrical resistance. In our system, PP was converted to amorphous carbon, which typically exhibits much higher electrical resistivity compared to metals, as well as their more graphitic counterparts. Therefore, under the same current, PP-derived carbon can reach a much higher temperature. Due to the lack of ability to on-demand produce carbon-only structures, previous work for making carbon-based Joule heaters largely focuses on polymer-carbon composites, which have a limited thermal stability as polymers can degrade when temperature is above 500 °C.

In our work, when the input voltage increased to ~25V, the 3D-printed carbon spring can increase temperature reaching ~810 °C. For the purpose of comparisons, most other reports for preparing Joule heaters cannot achieve such performance and requires sophisticated steps for materials synthesis. For example, graphene-wrapped sponge requires 55 V voltage to achieve ~400 °C (*Nature Nanotechnology*, **2017**, 12, 434), CoNC@GN/PCL/TPU composites needs 40 V to reach ~125 °C (*Advanced Functional Materials*, **2022**, DOI: 10.1002/adfm.202211352), and carbonized melamine foams reach ~200 °C under 7 V (*Journal of Materials Chemistry A*, **2021**, 9, 11268).

However, we are also concerned that an inaccurate comparison may be misleading. Below are key reasons: 1) As the temperature is determined by the total electrical resistance of the sample, it is difficult for different studies to compare against each other unless the sample has the exact same size and geometry. 2) With increasing Joule heating, the convection and radiation loss can be higher as well, which these values are also determined by sample shape and their surface area.

Therefore, in the revised manuscript, we included more discussions about Joule heating capability of our materials (focusing on why), but did not have direct comparisons as it is very challenging to conclude apples-to-apples comparison from different studies. The relevant discussion is shown below:

“Specifically, according to Joule’s law, increasing the electrical resistance of conductors can result in a higher temperature, which the amorphous nature of these PP-derived carbons provides a

significant advantage in the Joule heating capability compared to their counterparts of metals and more graphitic carbon-based, which are typically much more electrically conductive.”

“In conjugation with high thermal stability of carbons, this feature is particularly desired for the electrification of heating processes toward a carbon-neutral economy.”

6. RVE analysis is a bit confusing because the fiber is not clearly illustrated in Figure 7b and S22, and there is no clear description of the length scale (the voids are assumed to be the same diameter as CF, so $\sim 9 \mu\text{m}$?). It was also surprising that the isotropic assumption seems to work well enough even though the structure is highly anisotropic due to the FDM process.

Response: We appreciate the comments which have helped us improve the clarity of the RVE analysis. This comment has three questions, which we will answer one by one as follows:

Figure clarification: In Figure 7b (see below), the blue elements represent the carbon fibers, the grey elements represent the polymer-derived carbon matrix, and the white regions are embedded pores (no element). The caption of Figure 7 has been updated to clarify the color definition.

“Figure 7(b): RVE models for varied crosslinking time (grey: matrix; blue: carbon fibers; white: macropores).”

Additionally, we would like to further expand and clarify our simulation efforts to understand our 3D printed carbon systems.

Pore size effects: Yes, in most of our original models (except those with elliptical pores), the pores were assumed to have a circular shape with the same radius as the fibers. Informed by experiments, both fibers and pores were assumed to have a diameter of $10 \mu\text{m}$. To investigate how void size influences the mechanical behavior of composite RVE, we have done the following two studies:

- (a) In the original manuscript, we have studied how RVE performance could be influenced by the shape of macropores. The circular macropores were changed to elliptical pores with random orientations and various eccentricities, while the porosity was maintained at a constant. In addition to changing the shape, the pore size was also effectively modified because of the varied eccentricity:

1. Eccentricity = 0.0: major axis = 10 μm , minor axis = 10 μm
2. Eccentricity = 0.2: major axis = 10.1 μm , minor axis = 9.9 μm
3. Eccentricity = 0.4: major axis = 10.4 μm , minor axis = 9.6 μm
4. Eccentricity = 0.6: major axis = 11.2 μm , minor axis = 8.9 μm
5. Eccentricity = 0.8: major axis = 12.9 μm , minor axis = 7.8 μm
6. Eccentricity = 0.9: major axis = 15.1 μm , minor axis = 6.6 μm

The stress-strain curves of these samples indicate a very limited impact of the pore shape on mechanical properties. This provides us a preliminary conclusion that changing void sizes does not have a significant impact on the mechanical properties of our composites.

(b) In response to this reviewer comment, we performed additional computational work to investigate how void diameter influences mechanical properties. As shown below, we chose the RVE of 12 hours crosslinking time as the representative, which has a porosity of 5.8%. In this investigation, the void diameter was no longer a constant of 10 μm but randomized in the range of 3-17 μm . All voids have random locations. The fiber and void fractions were fixed.

The compressive and tensile strain stress curves clearly show that the randomized void size has negligible influence on the mechanical properties of RVE, given that the void volume fraction is unchanged.

Voids with random size

Voids with identical size

Figure caption: Stress-strain curves of PP-CF derived carbons with varied pore sizes in the matrix

Model anisotropy: We agree that FFF process prints material with anisotropy at different length scales. Our RVE model is simplified toward focusing on understanding in-plane mechanical properties of PP-CF derived carbons, which assumes (1) isotropic matrix and anisotropic fiber (different properties along fiber and along transverse directions; transversely isotropic); and (2) fibers with certain alignment. With these assumptions of anisotropic material and geometries, the RVE overall properties are anisotropic, which were further used in the macroscale analysis of the printed gyroid structure. As both experimental observation and computational simulation were focused on in-plane direction mechanical properties, our model can fit experimental data well. We agree that to understand mechanical properties along with the out-of-plane direction, the weakened interfaces between neighboring printing layers need to be fully considered. We believe this requires sophisticated investigations, which can be addressed in future studies.

Minor comments

1. Page 8, line 156: Easier to follow if Figure S5a is pointed out as a schematic for the reaction.

Response: We appreciate this suggestion. Following content is provided in the main text to address this comment.

“These results of indicating PP-CF crosslinking kinetics are consistent with gel fraction measurements (Figure S4) and Fourier transform infrared spectroscopy (FTIR) data (Figure S5); a simplified scheme illustrating chemical conversion of PP to crosslinked carbon precursor is also provided in Figure S5(a).”

2. Figure 7e: Error bars should be included.

Response: This is an important suggestion and error bars are now included in the new Figure 7e (see below). In response to this reviewer comment, we performed a significant amount (five replicates for each sample) of additional computation on RVEs with randomized fiber and void locations, which further enables a statistical analysis of the derived matrix properties including the error bars.

Reviewer #3 (Remarks to the Author):

This is a reviewer response to the manuscript titled “Lightweight and elastic carbons from 3D-printed plastics: accurate structural control enabling tunable mechanical properties.” The title is well-suited to the article as it describes a process of transforming polypropylene objects fabricated through fused filament fabrication (FFF) additive manufacturing (AM) into carbon. The titular “structural control” arises from varying the time allowed for sulfonation reaction (a post-print submersion in sulfuric acid). The authors demonstrate that removal from the sulfuric acid bath prior to what they term “complete sulfonation” yields unsaturated double bonds and uncrosslinked polypropylene that reduces the density of the final, carbonized structure through liberation of these moieties in the post-acid bath carbonizing process (high temperature pyrolysis in an inert atmosphere). The authors show that removing the parts from the acid bath at different times only affects the porosity of the structure without compromising their carbon yield. Therefore, compression properties can be tuned while maintaining the unique properties of carbon structures, including Joule heating, which the authors demonstrate. This manuscript extends the authors’ previously published work (Smith, et al. 2023; Reference [24]) through the inclusion of chopped carbon fiber (CF) in the PP filament. The evident difference in resulting carbon structure performance is noteworthy. The authors describe how including the carbon fibers aids in holding the printed structure together during the sulfonation, drying, and carbonization process. The increased stiffness and imbalance of sulfuric acid uptake results in oriented microcracks as the PP swells and CF does not. Although these cracks can be argued to diminish overall mechanical performance, the authors point out that without these cracks, full sulfonation (and therefore crosslinking) cannot occur since this is a diffusion limited process. Additionally, including CF significantly reduces shrinkage throughout the post-print chemical modification and conversion into carbon; this is a significant improvement over other published methods describing conversion of AM parts into carbon.

Response: We are very grateful for the reviewer reading our manuscript in detail and providing excellent summary and very valuable feedback. This clearly reflects the commitment and dedication of the reviewer, which is so important and much needed in the scientific community. We truly enjoyed reading and addressing the comments from this reviewer, which are very useful to further improve the quality our work.

The present work is highly impactful to a broad audience as the demonstrated means of converting an AM part to a carbon structure is accessible. The authors use commercially available off-the-shelf FFF machines (Ultimaker) and PP filament (Braskem), sulfuric acid is a common chemical, and the high-temperature furnace for carbonization is also commercially available, albeit likely the most expensive aspect of the set-up. Importantly, I believe I could replicate the results in my lab based on the information provided in the report. There is clearly much future work to be done in the characterization and application development for the described method – far more than a single research group can accomplish, which is why I recommend publication of this manuscript in Nature Communication.

Response: We particularly appreciate this comment from the reviewer, which highlights the broad and transformative impact of our work. Our system is indeed simple and scalable. We feel the same that many groups can be benefited from this report work to collectively advance fundamental and applied science toward future technology development in the area of AM of carbons. As this is the very first demonstration of 3D printing carbon with accurate structural control and modular mechanical properties, the authors are all very excited to see how this technology can be further used in different application domains.

While the submitted work is impactful and sufficient for publication (following the minor revisions suggested), it has left the Reviewer with a few thoughts and questions. Given the broad readership of Nature Communication, it is reasonable for these items to have been omitted from the submitted manuscript, but should certainly be addressed for a more complete depiction of the described process. The Reviewer notes the following topics of further discussion: (i) comparison of diffusion against other AM carbonization processes, (ii) the inseparable impact of part design on the carbonization process, and (iii) greater depth on the micro/mesoscale influence of the CF on carbonization. Naturally, the concerns of diffusion and design are linked and will be addressed together.

Response: We very much appreciate these major comments, as well as all details reviewer provided for improving our manuscript. We have now addressed all these comments through additional experiments and expanded discussions as shown below. We note while we did a lot of experiments and results are shown in the responses, some of them (regarding with processing parameters) are included in the revised manuscript with brief discussions. The discussions associated with mechanism and basic scientific parts are comprehensive. This is simply for the purpose of keeping this manuscript concise for readers to understand key concepts, preventing it to become too expansive. Since Nature Communication publishes response letter along with the

manuscript, results and discussion below will be available to the broad communities. We thank the reviewer for understanding us on this point.

Other publications describing carbonization of AM parts from polyimides involve an outgassing process inherent to the imidization reaction (e.g., Arrington, et al. 2021; Reference [12]). This process often results in catastrophic failure as the expanding gas generated from inside the bulk of the part creates large cracks as it escapes, similar to the combined burnout-sintering common to binder jetting (Rahman, Wei, Miyajima, and Williams Addit. Manuf. 2023) and fabricating ceramic structures through vat photopolymerization of highly loaded polymer resins (Cao, et al. Addit. Manuf. 2021). It seems that the process described in the present manuscript has exchanged the challenge of inside-to-outside diffusion of a gas for the outside-to-inside diffusion of the sulfuric acid. The authors state in the present work that the microcracks that form through the swelling strain mismatch between the CF and the PP matrix aid in deep penetration of sulfuric acid, and therefore thorough crosslinking throughout the printed PP structure. Additional work should understand the nature of the crosslinked network (e.g., molecular weight between crosslinks) and its dependence on sulfuric acid concentration and diffusion rate.

Response: We appreciate this comment. Two of the works the reviewer mentioned (from Rahman et al., and Cao et al.) involves the removal of polymer binders through thermal degradation to create ceramic structures with possible carbon residue; these polymer binders typically degraded to gaseous products and form many voids and pores upon carbonization. This is different from the goal of our work, which we converted PP to high char yield precursor for producing carbon parts through pyrolysis. The high carbon yield of crosslinked precursors ensures that limited amount of gaseous products would be produced upon carbonization, limiting their impact on disrupting printed parts (as the reviewer mentioned, the gas generated from inside the bulk of the part can create large cracks as it escapes), and thus all structures are completely retained with very low degree of shrinkage. The work from Arrington et al. is relevant, as their goal was also directly converting polymer precursors to carbons, which seems that they can also yield smooth monolith carbon products (as the SEM images shown in their Figure 3); in this case, ~10% dimensional shrinkage was observed from converting fully aromatic polyimide to carbonaceous product. Unfortunately, the mechanical properties were not available from this work (while the authors claim that they are mechanically robust).

In our system, cracks were formed due to reaction-induced stress, which facilitates the sulfuric acid diffusion and crosslinking of thicker parts. We agreed with the reviewer and have tried many attempts to understand the nature of PP crosslinked networks, but it has been experimentally very challenging to determine. First, crosslinking occurs through the radical-coupling induced formation of carbon-carbon bonds (a simplified reaction scheme is shown below) which makes it challenging to differentiate newly formed crosslinks between the chemical species that are already present in the precursor through techniques such as solid-state NMR (Nuclear Magnetic Resonance); the dynamic addition and elimination of sulfuric acid and other functional groups on the polymer backbones cannot be directly and accurately correlated to crosslinking degree.

There are other methods, including rheological experiments, which can be employed to determine crosslink density of polymer systems. We have tried to perform these additional experiments but 3D printed PP inherently has open-spaces (due to both in-fill density and inherent voids from FFF printing), which made quantitative dynamic mechanical analysis (DMA) measurement very challenging. We note there are a few other techniques for quantifying crosslink density based on rubber elasticity theory which have been established in the literature. One of which is based on the Flory-Rehner equation (*J Chem Phys*, 1950, **18**, 108-111) and the ability of a gel to uptake a compatible solvent (*Macromolecules*, 2022, **55**, 10900-10911). The crosslink density (n) is represented by the following equation:

$$n = \frac{-[\ln(1 - V_r) + V_r + \chi V_r^2]}{V_o (V_r^{\frac{1}{3}} - \frac{V_r}{2})}$$

where V_r is the volume fraction of the sample in the swollen state, χ is the interaction parameter between the solvent and the polymer, and V_o is the molar volume of the solvent. V_r can be calculated through the following equation based on the mass of the sample after swelling and the density of the solvent (m_1 , ρ_1), and the mass and density of the sample prior to swelling (m_2 , ρ_2).

$$V_r = \frac{\frac{m_2}{\rho_2}}{\frac{m_2}{\rho_2} + \frac{m_1 - m_2}{\rho_1}}$$

We attempted these experiments on the PP crosslinked samples using different solvents (including ethanol, toluene, water), but all swelling results are minimal and hard to detect. Additionally, as the reaction is spatially heterogenous within printed PP layers, these averaged measurements may lead to inaccuracy. In general, to the best of authors' knowledge, there has been no report demonstrating the capability of determining molecular weight between crosslinks

for sulfonation-induced, fully crosslinked polyolefin networks, likely due to all the reasons we just discussed.

However, we still agree that understanding the impact of reaction condition (particularly temperature and acid concentration) of PP crosslinking kinetics is important. Therefore, additional experiments were performed. First, using fuming sulfuric acid (~20% and contains free SO₃), the reaction kinetics became much faster due to the presence of free radicals to facilitate PP crosslinking. As shown below, using fuming acid the PP can be fully crosslinked after 2 h, which exhibit a plateau carbon yield value of 63 wt% after pyrolysis at 800 °C. This result suggests the opportunity to significantly reduce crosslinking time from 12 h (from using concentrated acid) to 2 h (from using free SO₃ containing fuming acid)

Figure caption: Change of polymer crystallinity of PP-CF parts as a function sulfonation time and their carbon yield using fuming acid and a crosslinking temperature of 150 °C.

Second, we varied reaction temperature to understand its impact on PP crosslinking kinetics, while addressing another comment from the reviewer about what the maximum temperature one could use for crosslinking of PP-CF. As shown below, with increasing reaction temperature from 150 °C to 170 °C, time required for fully crosslinking PP-CF reduces from 12 h to 2 h, indicating enhanced reaction kinetics, earlier crack initiation time, and faster acid diffusion rate. Further increasing the reaction temperature to 180 °C leads to major distortion of PP-CF samples, due to the gas byproducts from rigorous chemical reactions disrupting the framework.

Figure caption: (a) Images of PP-CF samples after sulfonation for 2 h at different temperature. Major distortion of samples was observed when the reaction temperature is 180 °C. (b) Crystallinity of PP-CF samples as a function of sulfonation time at different reaction temperature using concentrated acid.

Following discussion are now included in the revised manuscript:

“As an example, Figure S25 and S26 shows that by using fuming acid and/or increasing reaction temperature, faster PP-CF crosslinking kinetics can be obtained, resulting in significantly shorted reaction time to approximately 2 h. Specifically, the presence of free SO₃ group in the fuming acid can improve PP crosslinking kinetics, while higher sulfonation temperature can lead to enhanced reaction kinetics, earlier crack initiation time, and faster acid diffusion rate.”

All parts shown in the present manuscript are reported to have been printed at 20-40% infill; this allows for an ample surface area to volume ratio that limits the maximum distance into the part required for sulfuric acid penetration. The Reviewer wonders whether the described process could successfully create solid structures of carbon at any length scale, or what the maximum feature size might be. Fortunately, AM techniques excel at lattice-type structures so the issue of penetration depth may be mitigated through intelligent, science-base design; however, the Reviewer is not aware of the existence of any design guidelines for such an effort. In addition to “design for diffusion,” the effects of overall design choice cascade into carbonization and final mechanical properties. Although tailorable from a bulk sense, the mechanical properties of the final carbonized part have not been shown to be functionally graded, which is a desired ability in AM fabrication. Selective carbonization would be helpful for realizing so-called “smart structures” that combine structure and sensing capabilities. This does not seem possible with the currently presented process.

Response: We have now studied the impact of in-fill density of PP-CF parts on their crosslinking kinetics and dimensional shrinkage. As shown below, when increasing the in-fill density from 50% to 100%, the sulfonation time to fully crosslink PP matrix increases from 24 h to 72 h. These results suggest that in-fill design of printed parts has a strong impact to control acid diffusion kinetics, leading to altered overall sample crosslinking samples. Therefore, we believe that the reviewer made an excellent point on the need of “design for diffusion”, which AM based technologies are uniquely capable to accomplish. This also result may also suggest our process could create solid structures of carbon, while reaction condition need to be optimized and reaction time could be long.

Figure caption: Crystallinity of PP-CF as a function of sulfation time at 150 °C using concentrated acids with varied in-fill density of printed parts.

After carbonization, we found that the degree of in-plane shrinkage of these samples (fully crosslinked with respectively optimized crosslinking condition) was consistently low (~ 2%) and similar to our model systems with lower in-fill density, further confirming and the key concept of this work.

Figure caption: In-plane degree of shrinkage of PP-CF samples upon conversion from printed to carbonized states with varied in-fill density

Based on these results, several ideas are already emerged to leverage the different crosslinking kinetics of PP-CF as a function of their in-fill density. For example, can we crosslink a printed 3D printed PP-CF structure with gradient/graded in-fill density? Our hypothesis here is that upon the same reaction condition, the shrinkage of parts would remain to be very low, yet the spatially different properties can be developed within one sample, including both altered in-fill density and porosity, potentially leading to interesting mechanical properties and other performance. This may be a very interesting future work to pursue. We note, and agree with the reviewer that, this route may offer spatially selective carbonization, which can be helpful for realizing smart structures that combine structure and sensing capabilities.

Finally, the Reviewer would like to know more regarding the micro and mesoscale influence of the CF on the carbonization process. The current manuscript focuses on the stiffness of the already-carbonized CF and how that benefits the carbonization of the matrix PP throughout acid

swell and pyrolysis steps. It is unclear from the present manuscript if there is a templating effect of the CF either on the sulfonated PP or the final carbonized structure. Regardless of templating, it would be especially interesting to discover whether the alignment of carbons in the fiber regions is preserved following pyrolysis. If so, design (including toolpath planning) again becomes a critical feature.

Response: We agree that this is a templating effect from the presence of CFs, which significantly reduces the degree of shrinkage during PP carbonization process. In the manuscript, the authors expressed this concept using a different description: “We attribute very low dimensional shrinkage of our samples to the presence of CF in the printing filaments, which restricts the volumetric contraction of polymer matrix during their conversion to carbonaceous products. Similar filler impacts on significantly reducing volume shrinkage of precursors upon carbonization were also observed in several previous studies”. This discussion aligned with the reviewer’s thought.

In our previous work (*Adv Mater*, **2023**, 35(17), 2208029), we found that converting PP to carbons through pyrolysis, with the absence of CFs, led to ~20% shrinkage along the in-plane direction, and ~10% shrinkage along the out-of-plane direction. In addition, the impact of CF on degree of shrinkage of samples upon carbonization is shown below, further confirming our explanation.

In a recent study (*Adv Mater*, 2023, 35(38), 2208230), Fu et al., demonstrated that by containing 30 wt% carbon nanotubes in the polylactic acid (LCA) matrix, the removal of PLA matrix at high temperature pyrolysis only leads to less than 10% volumetric shrinkage, as shown below. The underlying mechanism is similar to our work.

Editorial Note: Top figure reproduced from Zhang, C., et al. Carbon additive manufacturing with a near-replica “green-to-brown” transformation. *Adv. Mater.* **35 (38)**, 2208230 (2023), with permission from John Wiley and Sons. <https://doi.org/10.1002/adma.202208230>

Additionally, we provided additional SEM images (below) in our revised manuscript to confirm that the alignment of CF is retained after carbonization, with additional discussions.

“the slight anisotropic shrinkage can be possibility due to the nature of FFF printed parts or the templating effect from the CFs, which are aligned with the in-plane direction. As shown in Figure S15, after carbonization the alignment of CF was still completely retained in the final carbon composites.”

Figure caption: SEM images of carbonized PP-CF samples confirming the alignment of CF in the matrix is preserved.

The Reviewer is admittedly unfamiliar with the field of carbonization; however, I estimate that carbons derived from PP should have a maximum yield of 86 % wt in relation to the starting mass of PP based on the weight ratio of carbon to hydrogen in the repeating unit. The authors state achieving 61% yield in the present work and provide two citations (one being the authors’ prior work; Reference [50]) to justify this value as typical. The Reviewer is curious where the balance of the mass goes; is it vaporized as carbon dioxide through a reaction with the sulfuric acid? Neither the current work nor the provided citations offer an explanation. Including an additional

reference that covers this rationale would enhance the value of the manuscript to a broader audience.

Response: While the number of references in converting PP to carbon through sulfonation is limited, there are several works confirming that the carbon yield of sulfonation-crosslinked PP is in the range of up to ~60%, including works from Guo et al., (*Science of The Total Environment*, **2022**, 817, 152995), Lee et al., (*Journal of Industrial and Engineering Chemistry*, **2022**, 105, 268), and Lin et al., (*Ionics*, **2021**, 27, 2169-2179); one of them is now included in the revised manuscript.

The crosslinking of PP is a free radical-based reaction, which is exothermic reaction due to the tertiary carbon in the polymer backbones and involves chain scissions. The functionalization/attachment of sulfuric acid groups on to the PP, which are subsequently abstracted to form unsaturated bonds in the polymer backbones, can also cause the loss of carbon molecules. We also performed the additional work (TGA-MS, thermogravimetric analysis-mass spectrometry) to study gaseous products from carbonization of PP, which SO_2 , CO and CO_2 can be found, confirming the loss of carbons and other atoms during this step. This information is provided below and included in the revised manuscript:

Figure caption: Thermogravimetric analysis-mass spectrometry confirms the presence of gaseous product during pyrolysis of crosslinked PP matrix, including SO_2 , CO_2 and CO molecules.

The Reviewer requests for the authors to include some discussion of annealing and crystallization as part of the discussion of sulfonation. The authors describe performing sulfonation at 150 °C, which although is below the observed peak melting temperature of PP (according to Figure S3), is certainly above its glass transition temperature. The authors state that a common issue in FFF parts is anisotropy due to layering for both strength and shrinkage. However, if held above glass transition temperature for multiple hours, one would expect some degree of macro-scale healing at the layer interface to alleviate this issue and promote increase isotropy. Often, annealing is impractical for FFF parts as it would induce slumping as these elevated temperatures. The Army Research Labs has reported on successful annealing of FFF printed objects using a multi-material

core/shell approach where the core remains structurally in-tact at the chosen annealing temperature while the lower temperature shell can reflow enough to increase isotropy and strengthen weld lines (Toal, Holmes, Rodriguez, and Wetzel Addit. Manuf. 2017). It is reasonable that a combination of the CF, the PP crystalline fraction, and the on-going crosslinking is able to provide the support at 150 °C to prevent slumping or macro-scale part distortion. It would be interesting to know the relation between crosslinking kinetics (i.e., rate) and rate/degree of healing of the weld interfaces as a function of temperature. Perhaps isotropy could be increased if the process began at a lower temperature for a longer time?

Response: We thank the reviewer for bringing up this excellent point. We have a similar thought when we first began this project, as our initial hypothesis included the sulfonation crosslinking can improve interlayer adhesion through these chemical reactions. However, our experimental results, such as from Figure S13, indicates that these local chemical crosslinking groups are still not sufficient to improve the isotropy in the macroscopic properties of printed parts. We believe this is because the crosslinking happens at the outside part of printed PP layers, which can result in kinetically trapped polymer chains with significantly hindered mobility to diffuse and weld the interfaces. Additionally, even with the absence of crosslinking, PP filament typically has high molecular weight, and these highly entangled polymer chains require a long time to diffuse across ~ mm regions. We thank the reviewer for a good suggestion of using core-shell filament to address this challenge, which can be pursued in future work. Similarly, there are several other approaches available to weld printed layers prior to sulfonation reaction, such as using microwave heating.

Following discussions have been included in our revised manuscript:

“Here, we note that even though the samples were thermally annealed at 150 °C for 24 h during sulfonation, significantly anisotropic mechanical properties were still observed in their mechanical properties (Figure S13), suggesting the crosslinking step alone can not efficiently weld the interfaces.”

“We attributed this result to several factors, including 1) commercial PP filaments are typically highly entangled and thus exhibit a slow diffusion rate, and 2) crosslinking of PP in acid occurs first at the interfaces between printed layers, which can kinetically trap the polymer chains and further hinder their ability to perform inter-layer diffusion for effectively welding.”

“Regarding with addressing the anisotropic mechanical properties, an inherent challenge from FFF method, we note several approaches can be employed to weld interfaces between printed layers of PP-CF, prior to the sulfonation reaction, such as microwave heating (cited work from Green et al.), or development of core-shell filaments (cited work from Wetzel et al. and Park et al.). For these structured filaments, during thermal treatment shell can easily flow to increase isotropy while the core remains structurally in-tact.”

In the Discussion section, the authors make statements that appear counter to what has been argued earlier in the manuscript. For instance, the Discussion section identifies “instrument cost,

process scalability, and accurate dimensional control” as “intractable” challenges. The Reviewer estimates the printer and filament used in this work to total < \$5,000. The authors themselves identify a key advantage of their process to be “low cost” (page 26). Which aspect of the process is cost prohibitive? On page 5, the authors report that the presented process is “simple and scalable.” Please explain how that can be true on page 5 but “scalability” is an intractable challenge on page 25. Finally, please clarify what would be an appropriate target tolerance for the intractable challenge of dimensional control. Especially please clarify whether this is a challenge unique to the presented carbonization process or else a more general challenge for the FFF modality of AM (which, for example, could be improved simply through using a more expensive, higher precision machine like a Stratasys instead of an Ultimaker).

Response: We are sorry for the confusion. The Discussion part of “instrument cost, process scalability, and accurate dimensional control” was identified as a previous significant challenge, which our work now is able to address. The total cost of entire process indeed is low, including using high commercially available recourses, which is less than a few thousand USD dollars; no element in our system and process is prohibitive. We have now revised our statement to improve clarity.

“However, most reported methods so far have several challenges that need to be addressed to enable AM of carbons at scale, including materials and instrument cost, process scalability, and accurate dimensional control over resulting parts after experiencing pyrolysis, hindering the scaled production and direct use of structured carbons in many important areas.”

To first clarify our point, the challenge from previous works is the almost unavoidable shrinkage of parts during carbonization process, which in this work it was approximately 2-4%; this value is already an order of magnitude higher than many other previous reports. Ideally, we would expect a zero-shrinkage, but of course it is very difficult as carbonization is able to condense the polymer matrix due to much higher density of carbon compared to polymers. The authors understand the reviewer’s comment of identifying what would be an appropriate target tolerance for structural control for AM of carbons, but providing a quantitative number here seems to be difficult (it would require a collaborative discussion between many groups as well as with industrial partners). However, we did mention the opportunity to address this challenge in our revised manuscript as shown below:

“This low shrinkage provides an opportunity to increase the manufacturing resource efficiency and product quality consistency, while the known shrinkage degree can be accommodated in the initial step of product design.”

Minor Revisions:

Given the broad readership of Nature Communications, it is understandable that the manuscript contains what are frequently used terms by the general public to refer to different AM technologies. However, the Review strongly encourages the authors to revise the manuscript to

include the standardized ASTM/ISO 52900 terms for AM technologies. This would include swapping the trademarked term “fused deposition modelling (FDM)” for the generic “fused filament fabrication (FFF)” or “filament material extrusion.” Additionally the manuscript references the legacy trademarked term “selective laser sintering,” which should be revised as “laser based powder bed fusion (PBF).”

Response: We thank these suggestions, and agree with the reviewer. The use of words has been changed to the standardized ASTM/ISO 52900 terms for AM technologies.

In a similar vein, please reserve “filament” for the feedstock form of the material and use the terms “layers” and “roads” to refer to material extruded during the printing process.

Response: We appreciate this feedback. We have now revised the manuscript accordingly to improve clarity as the reviewer indicated.

Please be sure that all references are correctly and consistently formatted in the Reference section. There is currently a mix of full author lists alongside “et al.” entries. Additionally, certain entries have inappropriate usage of italics or else are missing an appropriate usage. On page 14, the authors cite Reference [57] at the end of a sentence describing work published by “Liao, et al..” However, Reference [57] is not this work. Please ensure correct in-text citations for this and all other instances.

Response: The use of et al., is according to the policy of Nature Communication reference list style:

“All authors should be included in reference lists unless there are six or more, in which case only the first author should be given, followed by 'et al.'. Authors should be listed last name first, followed by a comma and initials (followed by full stops) of given names.”

We are very sorry for the mistake of References 57. We have now corrected it and confirmed all references are appropriate in this revised manuscript.

On page 15, there is a sentence with the phrase “...to prepare a variety of carbon materials containing complex geometries...” Is it not more accurate to state that the authors have prepared “a variety of complex geometries containing carbon materials?” As the Reviewer understands the work, the carbon produced does not vary (e.g., in D/G ratio) from part to part; only the porosity (i.e., a component of part geometry) is affected.

Response: We agreed with the reviewer for this suggestion and have made the correction accordingly.

“We employed our process to prepare a variety of carbon materials with different complex geometries, including a rhombic dodecahedral lattice, a golden eagle, a helmet, and a koi.”

On page 24, the authors use both the word “micropores” and “nanovoids” apparently referring to the same physical phenomenon. Please clarify which length scale most accurately represents these pore structures and revise accordingly.

Response: We appreciate this comment. We have now changed to nanovoids to micropores to improve the consistency and accuracy of our description.

REVIEWERS' COMMENTS

Reviewer #1 (Remarks to the Author):

The authors thoroughly addressed all comments/concerns, and this manuscript is now suitable for publication.

Reviewer #2 (Remarks to the Author):

I appreciate the authors' rigorous reassessment and additional work conducted. My former comments have been fully addressed, and thus, I recommend this manuscript for publication.

Reviewer #3 (Remarks to the Author):

The Reviewer thanks the authors for addressing the previous comments. The revised manuscript has been significantly improved. The Reviewer would like to clarify in the published revision notes one aspect from the discussion around the manuscript revision that was perhaps lost as the authors (thoroughly) addressed the other comments. As I see it, there are two main technologies employed to accomplish the described work: (1) filament material extrusion of cf-PP to create the macro-scale shape followed by (2) sulfonation + pyrolysis post-print chemical treatment to achieve a fully carbon structure retaining the overall shape set through 3D printing. Improvements to each of these two enabling technologies will improve the described result. Improvements might be realized in speed, throughput, scale, and precision in addition to many others. Additionally, there is nothing inherent to the described process that requires the AM process to be filament material extrusion; the only requirement is for the material to be PP (and preferably cf-PP). Therefore, some concerns on scale, precision, and isotropy could be addressed through preparing the printed structures via PBF as PP is also a commercially available material. Granted, this runs counter to the stated desire to keep the entire system low cost, as PBF systems are significantly more expensive than filament systems. Even still, there are other questions remaining regarding switching to PBF from filament extrusion on the extent and rate of diffusion into the printed part during the sulfonation step. The nature of each AM modality (i.e., filament extrusion vs PBF) results in a different form and extent of porosity – particularly surface-connected porosity. All this to say that although the steps of 3D printing and sulfonation + pyrolysis are in one sense independent and separate, they are also linked. An in-depth discussion of these differences is beyond the scope of the present work, but the Reviewer believes a short description should be included as part of the overall breadth of understanding regarding obtaining wholly-carbon structures from 3D printed parts. There is an interesting optimization problem here involving diffusion of liquid media into printed parts for post-print modification taking into account natural porosity, overall size, throughput, and cost. The filament extrusion system used in the presented work seems generally appropriate for the described sulfonation process.

Response to Reviewer's comment

Reviewer #3 (Remarks to the Author):

The Reviewer thanks the authors for addressing the previous comments. The revised manuscript has been significantly improved. The Reviewer would like to clarify in the published revision notes one aspect from the discussion around the manuscript revision that was perhaps lost as the authors (thoroughly) addressed the other comments. As I see it, there are two main technologies employed to accomplish the described work: (1) filament material extrusion of cf-PP to create the macro-scale shape followed by (2) sulfonation + pyrolysis post-print chemical treatment to achieve a fully carbon structure retaining the overall shape set through 3D printing. Improvements to each of these two enabling technologies will improve the described result. Improvements might be realized in speed, throughput, scale, and precision in addition to many others. Additionally, there is nothing inherent to the described process that requires the AM process to be filament material extrusion; the only requirement is for the material to be PP (and preferably cf-PP). Therefore, some concerns on scale, precision, and isotropy could be addressed through preparing the printed structures via PBF as PP is also a commercially available material. Granted, this runs counter to the stated desire to keep the entire system low cost, as PBF systems are significantly more expensive than filament systems. Even still, there are other questions remaining regarding switching to PBF from filament extrusion on the extent and rate of diffusion into the printed part during the sulfonation step. The nature of each AM modality (i.e., filament extrusion vs PBF) results in a different form and extent of porosity – particularly surface-connected porosity. All this to say that although the steps of 3D printing and sulfonation + pyrolysis are in one sense independent and separate, they are also linked. An in-depth discussion of these differences is beyond the scope of the present work, but the Reviewer believes a short description should be included as part of the overall breadth of understanding regarding obtaining wholly-carbon structures from 3D printed parts. There is an interesting optimization problem here involving diffusion of liquid media into printed parts for post-print modification taking into account natural porosity, overall size, throughput, and cost. The filament extrusion system used in the presented work seems generally appropriate for the described sulfonation process.

Response: We thank the reviewer for providing this critical comment and opportunity to further improve our manuscript. We agreed that the only requirement here is the use of polyolefin-precursors, and potential use of different 3D printing method to prepare PP and PP-CF parts, such as PBF-based, can lead to additional research opportunities for further strengthening the capability of 3D printing carbons. We have now included following discussions in our revised manuscript:

“Moreover, our approach of converting structured PP to carbons through steps of sulfonation and pyrolysis can be extended to different AM methods to prepare precursor parts, including a PBF-based method. The change in precursor preparations may result in parts with different void formation and layer thickness, and thus may lead to altered crosslinking and cracking kinetics. On this end, further studies can focus on understanding and optimizing the key material and process

parameters to control the diffusion of sulfuric acid into printed parts, while collectively considering factors of natural porosity, sample size, AM throughput, and system cost.”